**communications** engineering

# Numerical investigation of module-level inhomogeneous ageing in lithium-ion batteries from temperature gradients and electrical connection topologies
**Haosong He** [1], **Ashley Fly** [2], **Edward Barbour** [1] & **Xiangjie Chen** [1] ✉

The distribution of current/voltage can be further regulated by optimising the electrical connection topology, considering a particular battery thermal management systems. This study numerically investigates a 4P6S battery module with two connection topologies: 1) a straight connection topology, where the sub-modules consist of parallel-connected cells that are serial connected in a linear configuration, and 2) a parallelogram connection topology, where the sub-modules are serial connected in a parallelogram configuration. We find that the straight topology is more advantageous, as it allows the temperature gradient to be distributed among the parallel-connected cells in the sub-modules, mitigating over(dis)charging. Consequently, it achieves a 0.8% higher effective capacity than the parallelogram topology at 1C discharge, along with a higher state of health at 80.15% compared to 80% for the parallelogram topology. Notably, the straight topology results in a maximum current maldistribution of 0.24C at 1C discharge, which is considered an acceptable trade-off.

Cell ageing is a main concern in the electric vehicle (EV) industry, as it leads to performance degradation over time due to irreversible physical and chemical transformations. Cell ageing can be categorised into two types: calendar ageing and cycling ageing. Calendar ageing occurs predominantly when the battery is not in use, while cycling ageing happens when an EV is being charged or driven[1]. The calendar ageing rate increases with a higher state of charge (SOC) and operational temperatures, while it decreases over time due to the formation of the solid electrolyte interphase (SEI) layer, which develops proportionally to the inverse of square root of time[2]. Conversely, the rate of cycling ageing is influenced by several factors: it increases with the C-rate, SOC, and higher temperatures, and also accelerates under lower temperatures due to lithium plating[2,3].

In EVs, the battery pack comprises of a large number of cells (e.g. 444 18650 cells in a Tesla Model S module[4]), which can generate a substantial amount of heat during operation. Thus, an effective battery thermal management system (BTMS) is crucial in EVs to maintain uniform temperature distribution within the battery pack. Indirect liquid cooling has become the most prevalent BTMS in commercial EVs to date[5]. This method offers a quicker thermal response, higher cooling efficiency, and adaptability to both cooling and preheating conditions compared to forced air cooling[5]. When

compared to direct cooling (or immersion cooling), indirect cooling is favoured due to its design simplicity[6]. Moreover, the liquid coolant (e.g. ethylene glycol/water) has a lower viscosity than dielectric liquids (e.g. mineral oil), increasing higher flow rate with fixed pumping power[6]. Indirect cooling BTMSs typically use heat-conductive materials, such as cold plates and discrete tubes, to isolate cells from the cooling medium and also effectively channel heat away from the cells into the cooling medium. However, this design has a limitation: the thermal resistance from electrical insulating coatings between cells and the coolant reduces cooling efficiency. Additionally, commonly used coolants, such as water and glycol, are single-phase, which leads to an increase in coolant temperature along the cooling channel as heat is transferred from the cells. This process creates a temperature gradient within the battery modules, which is increased by higher C-rates and lower ambient temperatures, potentially resulting in uneven cell temperatures. Thus, BTMSs are required to limit the temperature gradient within the battery pack to a maximum of 5 °C[7].

The unavoidable temperature gradient between cells can also lead to cell-to-cell variations, which can be categorised into three levels: particle level, cell level, and module level. At the particle level, defects or irregularities in electrode materials due to manufacturing techniques can lead to local

[1]Centre for Renewable Energy Systems Technology (CREST), Wolfson School, Loughborough University, Holywell Park, Loughborough LE11 3GR Leicestershire, UK. [2]Department of Aeronautical and Automotive Engineering, Loughborough University, Loughborough LE11 3TU Leicestershire, UK. ✉e-mail: X.J.Chen@lboro.ac.uk

inhomogeneities, impacting the performance, durability, and safety of the battery[8]. However, particle-level inhomogeneities are typically intrinsic and difficult to control. To date, most studies focus on the cell level and explore aspects such as surface temperature inhomogeneities[9], capacity variation[10], internal resistance variation[10], and mechanical stress variation[11]. Research on module-level variations is relatively less explored. Previous module-level studies have predominantly focused on current regulation regarding thermal gradients[12,13], interconnection resistance[14,15], and welding techniques[14].

From a thermal perspective, temperature gradients lead to variances in temperature-sensitive electrochemical properties among cells, such as internal ohmic resistance, charge exchange current, and diffusion coefficient[16]. For example, ohmic resistance decreases with increasing temperature due to lithium ions migrating faster through the electrolyte[17]. The charge exchange current increases with temperature as electrodes become more reactive[18]. Similarly, the diffusion coefficient increases with temperature, enhancing the kinetics of lithium ions[19]. These variations lead to uneven distribution of temperature, current, and voltage, exacerbating inhomogeneous cycling ageing in the cells[7,20]. Liu et al.[13] found that a thermal gradient of 25 °C increased ageing rate by 5.2%. Various approaches have been proposed to regulate temperature homogeneity within the module, such as cell tab optimisation[21,22], low-temperature preheating/self-heating techniques[23], battery mechanical design[24], reconfigurable battery management systems[25], and novel balancing methods[26]. From the perspective of the BTMS, various aspects such as cell layouts[27], flow patterns[28], and BTMS geometries[29] have been studied as means to improve homogeneity.

Despite these optimisations providing more favourable operating conditions for cells, they offer limited flexibility once an optimal BTMS design is established. Therefore, with the constraint of a predetermined BTMS design, optimising the electrical connection topology emerges as an additional strategy to improve module performance. For example, Hosseinzadeh et al.[15] examined the short-term impact of cell-to-cell variations (i.e. interconnection resistance, thermal gradient, and capacity variations) and evaluated two distinct topologies - Z configuration and I configuration - on a 15P1S module. In the Z configuration, the positive and negative terminals were located at the two ends of the module, while in the I configuration, both terminals were placed on the same side of the module. They found that modules using the I configuration exhibited a stronger correlation with interconnection resistance, which was attributed to the cumulative effect of interconnection resistance for cells located further from the battery terminals. Conversely, the Z configuration reduced the amplification of interconnection resistance within a parallel system. Li et al.[30] also evaluated Z and I configurations in the context of an 11P1S module equipped with an air-cooling BTMS. They investigated different ageing rates under various parallel connection topologies with a fixed air-cooling BTMS position, finding that the I configuration with terminals on the air inlet side had the highest state of health (SOH) deviation (0.46%) after 400 cycles. In comparison, the I configuration with terminals on the opposite air inlet side had the lowest deviation (0.21%). The Z configuration had a deviation (0.33%), which lay between the two I configurations.

These aforementioned studies underscore that, under identical cooling conditions, inhomogeneous cycling ageing can be further managed by modifying electrical configurations. However, the limitations can be summarised as follows:

1. Most previous studies focus on parallel connections (nP1S), which may not adequately represent the current distribution at the module level.
2. Most previous studies rely on 2D models, which cannot fully capture real-time temperature changes or distribution throughout the entire cycling process. In these studies, the temperature gradient is often set to a constant value (e.g. a 5 °C increment in Liu et al.'s study[13]; a fixed gradient of 12.5 °C or 25 °C in Marlow et al.'s study[12]).
3. Most research has focussed on interconnection resistance and welding techniques to optimise module-level variances. The cell-to-cell variances caused by temperature gradients across different electrical connection topologies has not been reported.

Therefore, it is crucial to develop a 3D battery module model that includes both parallel and series connections while considering real-time temperature changes. This will enable a comprehensive understanding of module-level inhomogeneous cycling ageing caused by different electrical connection topologies. Specifically, this study will be underpinned by two distinct electrical connection topologies, that are adopted respectively by two mainstream EV manufacturers: Lucid and Tesla. For the Lucid design (see Fig. 1a[31]), the Model Air uses 30 21700 cells in a parallel configuration, assembled into 10 sub-modules (i.e. P1– P10; P1 is not shown in Fig. 1a) that are linearly connected in series, referred to herein as the 'straight connection topology'. Lucid employs end cooling, wherein the cooling plate contacts the bottom of the cell, with the coolant flowing from the top right side of the module and exiting on the top left side. For the Tesla design (see Fig. 1b[4]), the Model S incorporates 74 18650 cells in parallel, organised into 6 sub-modules (i.e. P1–P6) connected in series in a parallelogram configuration, referred to herein as the 'parallelogram connection topology'. Tesla utilises side cooling, where a serpentine cooling pipe wraps around the cell curved surface, with coolant flowing from the top side of the module and exiting on the bottom side. These different configurations provide a valuable reference for investigating the influence of electrical connection topology on the inhomogeneous cycling ageing.

This study numerically investigates the impact of the straight and parallelogram connection topologies on inhomogeneous cycling ageing due to temperature gradients stemming from BTMS by comparing the current, voltage, effective capacity, and SOH. The study is conducted in the following steps: 1) electrochemical and ageing parameters, introduced in Section 'Model development', are fitted based on experimental cycling and ageing data, as described in Section 'Parameterisation'. 2) 2D models configured in 1P3S and 3P1S are developed to examine current/voltage maldistribution (Section 'Influence of temperature gradient on electrochemical parameters') and inhomogeneous cycling ageing (Section 'Influence of temperature gradient on ageing rate') under a fixed temperature difference at sub-module level. 3) Since a 2D model cannot fully reflect real-time temperature changes, a representative 3D battery module (4P6S) with two different electrical connection topologies (i.e. the straight connection topology and the parallelogram connection topology) equipped with the same BTMS, is proposed to explore the inhomogeneous cycling ageing at module level. This module maintains the same module-level voltage as the Tesla Model S module but with reduced capacity. It reflects the key features of each electrical connection topology in a more compact size. The proposed 3D model offers a more accurate simulation of real-time temperature changes,

**Fig. 1 | Two different electrical connection topologies of parallel-connected sub-modules 1–10 (P1–P10) in electric vehicle modules. a** The straight connection topology of P1–P10: Lucid Model Air battery module. **b** The parallelogram connection topology of P1–P6: Tesla Model S battery module.

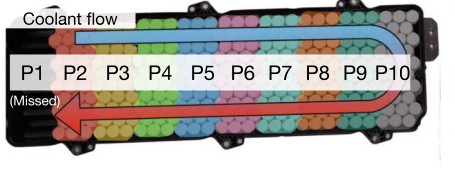
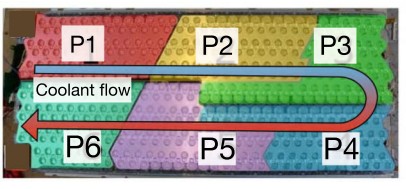

(a) (b)

enabling a thorough investigation of current/voltage maldistribution (Section 'Module-level current, voltage, and temperature distributions) and inhomogeneous cycling ageing (Section 'Module-level inhomogeneous cycling ageing'). Ultimately, this study advances the understanding of battery pack topology optimisation.

We find that the straight topology is more advantageous, as it allows the temperature gradient to be distributed among the parallel-connected cells in the sub-modules, mitigating the over(dis)charging issue. This results in a higher effective capacity increase of 0.8% than the parallelogram topology at 1C discharge. Additionally, it exhibits a higher SOH of 80.15%, compared to 80% for the parallelogram connection topology. However, the straight topology results in a maximum current maldistribution of 0.24C within sub-modules at 1C discharge, compared to 0.05C for the parallelogram topology, which is considered an acceptable trade-off.

## Methods
### Geometry development
**2D model**. The 2D battery modules configured in 1P3S and 3P1S, as illustrated in Fig. 2a, b, are built to investigate the impact of temperature variations on cells with a fixed temperature gradient at sub-module level. The temperatures of the cells are arbitrarily set to constants of 15 °C, 25 °C, and 35 °C, respectively. In the 1P3S configuration, cells are operated under a consistent potential, whereas in the 3P1S configuration, cells operate under a consistent current.

**3D Model**. 2D models explore the inhomogeneous cycling ageing under a constant temperature difference at sub-module level. However, heat generation from cells varies during the cycling phase, indicating that the temperature difference changes during an EV's operation[4]. Thus, a representative 3D battery module (4P6S) with two different electrical connection topologies (i.e. the straight connection topology and parallelogram connection topology) is proposed to investigate and estimate the real-time changes in temperature, current, and voltage of the cells at module level. The schematic diagrams of electrical connections for the straight connection topology and parallelogram connection topology are illustrated in Fig. 3a and b, respectively. Each sub-module is labelled from P1 to P6. The top view of the two electrical connection topologies and the flow direction of the coolant are illustrated in Fig. 3c, d. In the straight connection topology, parallel-connected cells are arranged in a line and stacked linearly in series (see Fig. 3c), while in the parallelogram connection topology, cells are arranged in a block and connected in a parallelogram shape in series (see Fig. 3d). The detailed 3D geometric dimensions of the straight connection topology and parallelogram connection topology are illustrated in Fig. 3e and f, respectively. Finally, each cell is labelled based on its position, as illustrated in Fig. 3g. For example, sub-module P1 contains $Cell_{1,1}$, $Cell_{2,1}$, $Cell_{3,1}$, and $Cell_{4,1}$ in the straight connection topology, and $Cell_{1,1}$, $Cell_{1,2}$, $Cell_{2,1}$, and $Cell_{2,2}$ in the parallelogram connection topology.

A Panasonic NCR18650PF lithium-ion battery with nickel cobalt aluminium oxide (NCA) anode and graphite cathode cell is used in this study. The specification of NCR18650PF is listed in Table S1. The reference

temperature of this study is 25 °C. The proposed multiphysics models are developed in COMSOL Multiphysics 6.2 (COMSOL). Before the numerical simulation, the following assumptions have been made in this study:

1. Cell-to-cell variances due to intrinsic reasons, such as the cell capacity, internal resistance, and energy density, are not considered as this study focuses on the cell-to-cell variance in temperature-sensitive electrochemical parameters due to temperature gradients (i.e. ohmic resistance, exchange current, and diffusion coefficient)[4,16]. Therefore, cells are assumed to exhibit the same electrochemical properties when they are at the same temperature.

2. Cell-to-cell variances due to extrinsic factors, such as interconnection resistance and welding techniques, are not accounted for in controlling the variables. This is to focus on the impact of connection topology on inhomogeneous ageing due to temperature gradients. Thus, interconnection resistance is neglected in the 2D model. In the 3D model, the electrical conductivity of connectors is manually set to an extremely high value of $1 \times 10^{12}$ S m$^{-1}$ to minimise the voltage drop and ohmic heating across the connectors. The influence of interconnection resistance is further discussed in Section 'Module-level current, voltage, and temperature distributions'.

3. The temperature-related ageing rate varies under cycling or resting conditions, and increases at lower temperatures during cycling[32], but decreases at lower temperatures during calendar ageing[33]. This study exclusively focuses on cyclic ageing due to cells experiencing the same calendar ageing rate when the temperature distribution is homogeneous and the cell is resting.

4. The ageing mechanism considered in this study is the SEI formation, which is the main ageing process in most graphite-based lithium-ion batteries[3]. Lithium plating is not considered since it mainly occurs under low temperature or high C-rate conditions[3].

5. Cell ohmic resistance changes due to ageing are not considered for the simplification purpose. The influence of internal resistance change is further discussed in Section 'Module-level current, voltage, and temperature distributions'.

6. The heat transfer considered in this model is limited to heat convection (i.e. heat removed by the cooling liquid) and heat conduction (i.e. heat transfer from the cells to the cooling pipe and between cells through the busbar). Heat transfer from cell to cell and from cell to ambient air is not considered. We note cells tend to have an insulation layer in battery pack design to prevent thermal runaway propagation[34,35].

**The LSPM**. To estimate the battery behaviours under different temperatures, Ekström et al.[16] proposed a LSPM. This makes an assumption that only one electrode contributes to diffusion-related voltage losses at the cell level, thus enabling the replacement of two electrode particles in the original SPM with a single particle. They employed the Levenberg-Marquardt (LM) optimisation algorithm to calibrate the electrochemical parameters to fit experimental data. The detailed exposition of LM algorithm can be found in ref.[36]. Furthermore, by integrating the Arrhenius equation with temperature-

**Fig. 2 | Schematic of electric connections for 2D lithium-ion battery (LIB) models. a** The serial connection (1P3S). **b** The parallel connection (3P1S) with a fixed temperature gradient of 10 °C.

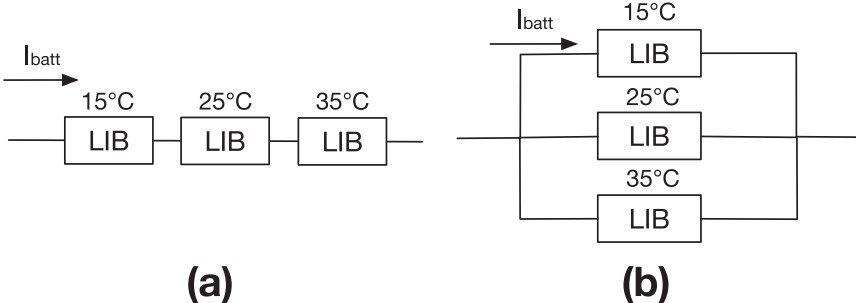

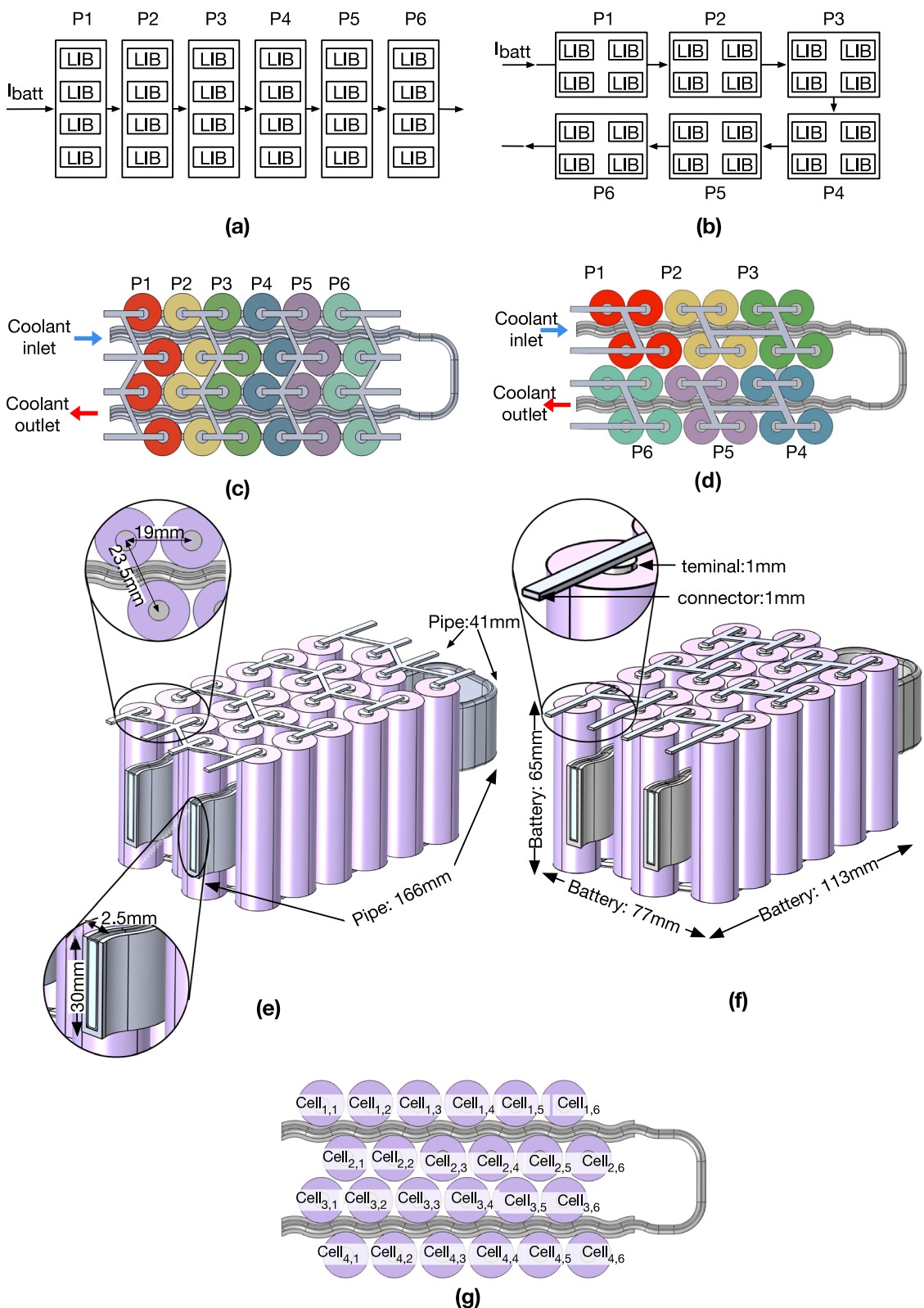

**Fig. 3 | Schematic of electric connections for 3D lithium-ion battery (LIB) modules (P1–P6 refers to the number of parallel-connected sub-modules).** **a** Straight connection topology based on the Lucid design, and **b** parallelogram connection topology based on the Tesla design. The top view of electrical connection: **c** straight connection topology, and **d** parallelogram connection topology. The 3D geometry of: **e** straight connection topology, and **f** parallelogram connection topology. **g** The cell labels.

sensitive electrochemical parameters, the LSPM can estimate battery behaviours under different temperature conditions.

The total overpotential is pertinent to ohmic, activation, and concentration overpotential, which relates to the irreversible heat in cell and given by:

$$\eta = \eta_{ohm} + \eta_{act} + \eta_{conc} \qquad (1)$$

Here, the ohmic overpotential ($\eta_{ohm}$) is due to ohmic resistance ($R_{ohm}$) in the movement of ions and electrons, as described by Ohm's law, given by:

$$\eta_{ohm} = R_{ohm}I_{batt} = \frac{\eta_{ohm,1C}}{I_{1C}}I_{batt}, \qquad (2)$$

where $I_{batt}$ is the applied current, $I_{1C}$ is 1C current, $\eta_{ohm,1C}$ is the ohmic overpotential at 1C.

The activation overpotential ($\eta_{act}$) reflects the activation barrier in the electrochemical reactions taking place in the cell, given by:

$$\eta_{act} = \frac{2RT}{F}\text{arcsinh}\left(\frac{I_{batt}}{2J_0I_{1C}}\right), \qquad (3)$$

where $R$ is the molar ideal gas constant, $F$ is the Faraday constant, T is the absolute temperature of cell, $J_0$ is the dimensionless charge exchange current rate.

The concentration overpotential ($\eta_{conc}$) is caused by the finite transport rates of reacting material within the cell, and can be calculated by applying Fick's diffusion equation to a dimensionless local particle SOC (S) over a one-dimensional particle length of 1. The equation incorporates a dimensionless spatial variable X, which varies from 1 (particle surface) to 0 (particle centre), given by:

$$\tau\frac{dS}{dt} = \nabla^2S, \qquad (4)$$

where $\tau$ is the diffusion time constant. The boundary conditions are given by:

$$\nabla S|_{X=0} = 0 \qquad (5)$$

$$\nabla S|_{X=1} = -\frac{\tau I_{batt}}{N_{shape}Q_{batt,0}}, \qquad (6)$$

where $N_{shape}$ is 3 for the spherical coordinate in this study, $Q_{batt,0}$ is the initial cell capacity.

The battery average SOC ($SOC_{ave}$) is obtained by integrating over the volume of the particle, is given as:

$$SOC_{ave} = 3\int_0^1 S^2dX \qquad (7)$$

Therefore, the concentration overpotential ($\eta_{conc}$) is given by:

$$\eta_{conc} = E_{OCV}(S|_{X=1}, T) - E_{OCV}(SOC_{ave}, T) \qquad (8)$$

The entropy change of the cell is given by:

$$E_{entropy} = (T - T_{ref})\frac{\partial E_{OCV}(SOC_{ave})}{\partial T}, \qquad (9)$$

where the entropy coefficient of the Panasonic NCR18650PF cell is taken from ref. 37, and listed in in Supplementary Table S2.

Thus, the OCV under various temperatures ($E_{OCV}(SOC_{ave}, T)$) is the sum of reference OCV ($E_{OCV,ref}(SOC_{ave})$) and the entropy voltage ($E_{entropy}$)

change, given by:

$$E_{OCV}(SOC_{ave}, T) = E_{OCV,ref}(SOC_{ave}) + E_{entropy} \qquad (10)$$

The terminal voltage ($E_{batt}$) can be calculated by the sum of the overpotential and the OCV, given by

$$E_{batt} = E_{OCV,ref}(SOC_{ave}) + E_{entropy} + \eta_{ohm} + \eta_{act} + \eta_{conc} \qquad (11)$$

**Temperature.** the electrochemical parameters ($\eta_{ohm,1C}$, $J_0$ and $\tau$) applied in LSPM are temperature sensitive, which can be modelled by coupling the Arrhenius dependency[16]. Arrhenius dependency relates the temperature sensitive parameters of the cell at present temperature to that parameters' values at a reference temperature (i.e. 25 °C in this study) via an exponential function with activation energy $E_{act}$[16], given by:

$$\eta_{ohm,1C} = \eta_{ohm,1C,ref}\;exp\left[\frac{E_{act}^{\eta_{ohm,1C}}}{R}\left(\frac{1}{T} - \frac{1}{T_{ref}}\right)\right] \qquad (12)$$

$$J_0 = J_{0,ref}\;exp\left[\frac{E_{act}^{J_0}}{R}\left(\frac{1}{T} - \frac{1}{T_{ref}}\right)\right] \qquad (13)$$

$$\tau = \tau_{ref}\;exp\left[\frac{E_{act}^{\tau}}{R}\left(\frac{1}{T} - \frac{1}{T_{ref}}\right)\right], \qquad (14)$$

where $\eta_{ohm,1C,ref}$, $J_{0,ref}$ and $\tau_{ref}$ represent their values under the reference temperature.

**Ageing.** For the ageing prediction, a semi-empirical ageing model is introduced into the LSPM[38], which considers the SEI formation as the main cause of battery degradation, whereby a parasitic loss current ($I_{loss}$) is utilised to compute the capacity loss of the cell due to side reactions forming the SEI layer on the graphite particles[38,39]. This parasitic loss current represents the sum of current contributions from the surfaces of the graphite particles that are covered by the intact microporous SEI layer, and the SEI layer that has cracked due to the expansion of the graphite particles. The parasitic loss current is affected by calendar time, cell voltage, current, ageing history, and temperature.

The capacity loss ($Q_{loss}$) can be described by an integration of the parasitic SEI current over time, given by:

$$Q_{loss} = \int I_{loss}\partial t \qquad (15)$$

Here, $I_{loss}$ is given by:

$$I_{loss} = \frac{Q_{cell,0}}{\tau_{loss}}f, \qquad (16)$$

where $\tau_{loss}$ is a calendar ageing time constant defining the rate of the parasitic reactions, and f is the ageing factor. Typically, capacity loss is associated with the cell voltage, the capacity throughput, ageing history and temperature, described by the dimensionless ageing factors: $f_E$, $f_I$, $f_{aged}$, $f_T$, respectively[39], given by:

$$f = f_Ef_If_{aged}f_T \qquad (17)$$

High SOC values (typically resulting in high battery voltage) accelerate capacity loss[38]. $f_E$ represents the result of either a parasitic electrochemical reduction reaction occurring on the negative electrode, or an oxidation

reaction occurring on the positive electrode, given by:

$$f_E = \exp\left(\frac{\alpha F\left(E_{OCV}(SOC, T) + \eta_{act} - \overline{E_{OCV}}(T) - E_{offset}\right)}{RT}\right) \quad (18)$$

Here, the transfer coefficient ($\alpha$) and offset potential ($E_{offset}$) parameters relate how the rate of the parasitic reactions changes when the battery voltage deviates from the average open circuit voltage ($\overline{E_{OCV}}(T)$), given by:

$$\overline{E_{OCV}}(T) = \int_0^1 E_{OCV}(SOC, T)\partial SOC \quad (19)$$

Battery lifetime relates to the amount of equivalent full cycles. $f_I$ represents the linear relation between the capacity fade and the number of full cycles, given by:

$$f_I = 1 + H\frac{\tau_{loss}|I_{batt}|}{2Q_{cell,0}}, \quad (20)$$

where H is cycling capacity loss factor, representing the additional (dimensionless) capacity loss induced by cycling.

Ageing history defines how many times the capacity loss rate will have been reduced when all capacity has been lost. $f_{aged}$ represents the rate of the capacity fade slowed down by the parasitic reactions, e.g. the formation of SEI, given by:

$$f_{aged} = \frac{1}{1 + (G - 1)\frac{Q_{loss}}{Q_{cell,0}}}, \quad (21)$$

where G is decelerating ageing factor, representing how many times the capacity fade rate has been reduced when all capacity has been lost.

The cycling ageing rate increase in both higher and lower temperature[40]. $f_T$ is the ageing caused by temperature, described by an Arrhenius expression:

$$f_T = \exp\left(-\frac{E_{act}}{R}\left(\frac{1}{T} - \frac{1}{T_{ref}}\right)\right), \quad (22)$$

where $E_{act}$ is the activation energy.

The symbols and values used in this section are listed in Supplementary Table S3.

**Heat generation rate**. According to Bernardi's heat generation model[41] total heat generation rate ($\dot{Q}_{total}$) from cell consists of the irreversible heat ($\dot{Q}_{irrev}$) and the reversible heat ($\dot{Q}_{rev}$).

$$\dot{Q}_{total} = \dot{Q}_{irrev} + \dot{Q}_{rev} \quad (23)$$

$\dot{Q}_{irrev}$ is caused by[39]: 1) Ohmic losses in the electrolyte, electrodes, and current collectors ($\dot{Q}_{\eta_{ohm}}$), 2) activation losses for the charge transfer reactions ($\dot{Q}_{\eta_{act}}$), and 3) concentration losses resulting from concentration gradients that generate heat due to non-ideal heat of mixing ($\dot{Q}_{mix}$), given by:

$$\dot{Q}_{irrev} = \dot{Q}_{\eta_{ohm}} + \dot{Q}_{\eta_{act}} + \dot{Q}_{mix}, \quad (24)$$

where:

$$\dot{Q}_{\eta_{ohm}} = \eta_{ohm}I_{batt} \quad (25)$$

$$\dot{Q}_{\eta_{act}} = \eta_{act}I_{batt} \quad (26)$$

$$\dot{Q}_{mix} = \frac{N_{shape}Q_{cell,0}}{\tau_{Arrh}}\int_0^1 \frac{\partial E_{OCV,therm}}{\partial S}\frac{\partial^2 S}{\partial^2 X}X^{N_{shape}-1}\partial X \quad (27)$$

Here, the thermo neutral voltage is given by:

$$E_{OCV,therm} = E_{OCV,ref}(S) - T_{ref}\frac{\partial E_{OCV}(S)}{\partial T} \quad (28)$$

$\dot{Q}_{rev}$ is caused by the entropy changes in the electrode reactions, given by:

$$\dot{Q}_{rev} = T\frac{\partial E_{OCV}(S|_{x=1})}{\partial T}I_{batt} \quad (29)$$

**Heat transfer**. An aluminium cooling pipe is wrapped around the battery module, and steel busbar connectors are positioned at the top and bottom terminals of the cells. To cool down the cells, water flows through the cooling pipe. The module's surface is assumed to be adiabatic, meaning the heat transfer occurring through the air is ignored.

Deionised water is used as the cooling fluid inside the cooling channels. The energy equation for water is given by:

$$\rho_{cool}C_{p,cool}[\partial T_{cool}/\partial t + (\vec{v} \cdot \nabla)T_{cool}] = k_{cool}\nabla^2 T_{cool}, \quad (30)$$

where the $\rho_{cool}$, $C_{p,cool}$, $T_{cool}$, and $k_{cool}$ is the density, heat capacity, temperature and thermal conductivity of water, respectively.

The motion of incompressible deionised water is governed by the mass conservation equation, given by:

$$\partial \rho_{cool}/\partial t = -\nabla(\vec{v})\rho_{cool} \quad (31)$$

and momentum conservation equation, given by:

$$\rho_{cool}(\partial \vec{v}/\partial t + (\vec{v}\nabla)\vec{v}) = -\nabla P, \quad (32)$$

where the $\vec{v}$ is the velocity vector of the water.

The energy conservation equation of cells is given by:

$$\rho_{batt}C_{p,batt}\partial T_{batt}/\partial t = \nabla(k_{batt}\nabla T_{batt}) + \dot{Q}_{total}, \quad (33)$$

where the $\rho_{batt}$, $C_{p,batt}$, $T_{batt}$, and $k_{batt}$ is the density, heat capacity, temperature and thermal conductivity of cell, respectively. Notably, thermal conductivity of cell is $k_{batt,r}$ and $k_{batt,a}$ for the radial and axial direction of cell, respectively.

The energy conservation equation of cooling pipe is given by:

$$\rho_{pipe}C_{p,pipe}\partial T_{pipe}/\partial t = \nabla(k_{pipe}\nabla T_{pipe}), \quad (34)$$

where the $\rho_{pipe}$, $C_{p,pipe}$, $T_{pipe}$, and $k_{pipe}$ is the density, heat capacity, temperature and thermal conductivity of pipe (i.e. aluminium), respectively.

The energy conservation equation of busbar is given by:

$$\rho_{bus}C_{p,bus}\partial T_{bus}/\partial t = \nabla(k_{bus}\nabla T_{bus}), \quad (35)$$

where the $\rho_{bus}$, $C_{p,bus}$, $T_{bus}$, and $k_{bus}$ is the density, heat capacity, temperature and thermal conductivity of busbar (i.e. steel AISI 4340), respectively.

The detailed values of each component in the battery module are provided in Supplementary Table S4.

## Results and discussion
### Parameterisation
The fitting process can be divided into two primary steps: 1) fitting of the electrochemical parameters, and 2) fitting of the ageing parameters. A 5000 s random mix of US06, Urban Dynamometer Driving Schedule (UDDS), Highway Fuel Economy Driving Schedule (HWFET), LA92, and Neural Network drive cycle data conducted on a single Panasonic 18650PF cell by Kollmeyer et al.[42] with a starting temperature of 25 °C, is utilised in the fitting process. The Fast Activation Energy Derivation (FAED) optimisation

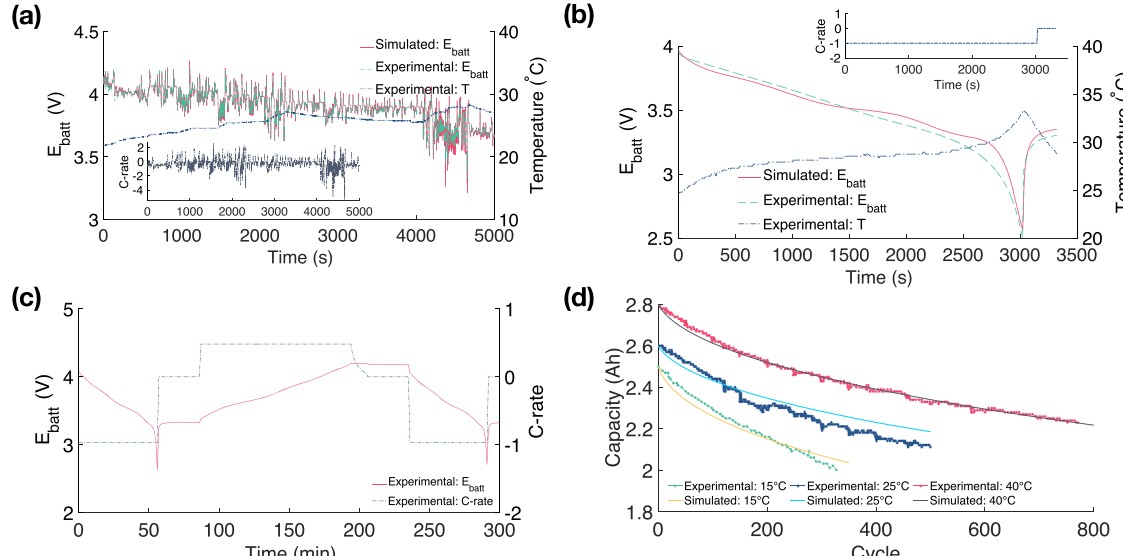

**Fig. 4 | The electrochemical and ageing parameterisations. a** The comparison between the experimental and simulated voltage for a drive cycle from 0 to 5000 s. **b** Verification of the accuracy of the fitted electrochemical parameters at 1C discharge. **c** Ageing duty cycle profile: C-rate and voltage change. **d** Experimental and simulated battery capacity degradation at 10 °C, 25 °C, and 40 °C; the ageing parameters are fitted experimental data of 10 °C and 40 °C, verified at 25 °C.

approach based on the least square fitting proposed by He et al.[4] is adopted for the electrochemical parameters fitting. Figure 4a displays the comparison between the simulated and experimental module voltages across various C-rates (ranging from −5.5C to 2.5C) and temperatures (ranging from 22 °C to 28 °C). The mean relative error for this comparison stands at 0.47%. The fitted electrochemical parameters can be found in Table 1. To validate the accuracy of the fitted parameters, 1C discharge data is employed, with temperatures ranging from 24 °C to 34 °C. Figure 4b illustrates the comparison between the simulated and experimental cell voltage, resulting in mean relative errors of 1.5%.

The ageing parameters are fitted and verified based on the Panasonic 18650PF capacity-cycle data from Zhang et al.[43] at three distinct temperatures: 10 °C, 25 °C, and 40 °C. Figure 4c illustrates the ageing cycling profile: the cell is discharged at 1C until it reaches the cut-off voltage of 2.5 V, followed by a rest period of 30 min. Then, it is charged at a constant current (CC) of 0.5C, transitioning to a constant voltage (CV) of 4.2 V with a cut-off current of 0.05C, and rests for another 30 min. Figure 4d illustrates the experimental and simulated ageing rates at 10 °C, 25 °C, and 40 °C. The LM algorithm is adopted in the ageing parameters fitting. Ageing parameters are fitted based on the ageing data at 40 °C. Notably, cell ageing exhibits different rates in different temperature ranges. Waldmann et al.[40] found that the slowest ageing rate is observed at 25 °C, while the ageing rates both increase at higher/lower temperature ranges (i.e. −20 °C to 25 °C and 25 °C to 70 °C). In this study, therefore, $E_a$ is fitted separately with the ageing data at 10 °C to represent the temperature range below 25 °C. The fitted ageing parameters are presented in Table 1. Lastly, the fitted ageing parameters are verified at 25 °C, and the fitted and experimental ageing results at 10 °C,

25 °C, and 40 °C are compared, with mean relative errors of 1.03%, 1.79%, and 0.40%, respectively.

## Influence of temperature gradient on electrochemical parameters

The temperature sensitive electrochemical parameters (i.e. $R_{ohm}$, $J_0$, and $\tau$) are the main factors leading to the current and voltage maldistribution when temperature distributes inhomogeneously. Figure 5 compares the variations in electrochemical parameters due to temperature differences for the serial and parallel connections. The duty cycle is the same as the ageing test: a CC charge at 0.5C, followed by a CV of 4.2 V, with a cut-off current rate of 0.05C, before resting for 30 min and being discharged under a current rate of 1C until the 2.5 V per cell cut-off voltage is reached, followed by another 30 min rest. It can be observed that cells connected in series share the same current, as illustrated in Fig. 5a, while cells connected in parallel experience different currents as they share the same voltage, as illustrated in Fig. 5b.

Figure 5c illustrates $R_{ohm}$ values of 25.1 mΩ, 17.8 mΩ, and 12.9 mΩ at 10 °C, 25 °C, and 35 °C, respectively, aligning with the theory that ohmic resistance decreases as temperature increases[44]. It is observed that a 20 °C temperature increase results in an approximately −0.61 mΩ°C⁻¹ change in $R_{ohm}$. This change aligns with the study by Wildfeuer et al.[45], which reports ~20% reduction in resistance from 25 °C to 35 °C. As suggested by Eq. (2), there is a linear relationship between $\eta_{ohm}$ and $R_{ohm}$, indicating that $\eta_{ohm}$ aligns with $R_{ohm}$. For serial-connected cells, the applied current is the same, leading to a 27% decrease in $\eta_{ohm}$, as illustrated in Fig. 5d. For parallel-connected cells, where the applied voltage remains constant, the cell with a higher $R_{ohm}$ will experience a lower current and subsequently a lower $\eta_{ohm}$, as depicted in Fig. 5e.

Figure 5f illustrates that the $J_0$ values are 0.05, 0.29, and 1.39 at 10 °C, 25 °C, and 35 °C, respectively. Lower exchange current denotes slower electrode reaction rates[46], which subsequently lead to higher $\eta_{act}$ values under identical applied currents. Figure 5g further illustrates this point, showing $\eta_{act}$ values of 102.1 mV, 66.5 mV, and 38.0 mV recorded at 10 °C, 25 °C, and 35 °C, respectively under a 1C discharge for serial-connected cells. For serial-connected cells, there is approximately 62% decrease in $\eta_{act}$. In contrast, for parallel connections, the applied voltage remains constant for each cell, implying that the cell with the higher electrode reaction rates will carry a lower current, and subsequently exhibit a lower $\eta_{act}$, as illustrated in Fig. 5h.

## Table 1 | The optimisation results of the electrochemical and ageing parameters

**Electrochemical parameters:**

| $\eta_{ohm,1C}$ (mV) | $J_0$(-) | $\tau$ (s) | $E_{act}^{\eta_{ohm,1C}}$ (kJ mol⁻¹) | $E_{act}^{J_0}$ (kJ mol⁻¹) | $E_{act}^{\tau}$ (kJ mol⁻¹) |
|---|---|---|---|---|---|
| 51.7 | 0.3 | 6712 | 24.36 | −58.98 | 18.58 |

Ageing parameters:

| $\tau_{loss}$ (s) | $E_{offset}$ (V) | $\alpha$ (-) | G (-) | H (-) | $E_a$ (kJ mol⁻¹) |
|---|---|---|---|---|---|
| $1.5812 \times 10^8$ | 0.36 | 0.31 | 49.67 | $2.14 \times 10^{-3}$ | −84.6 (10–25 °C) 56.23 (25–45 °C) |

**Fig. 5 | Comparison of electrochemical parameters for serial and parallel connections.** The variances in (**a**, **b**) current, (**c**) ohmic resistance, (**d**, **e**) ohmic overpotential, (**f**) dimensionless exchange current, (**g**, **h**) activation overpotential, (**i**) time constant and (**j**, **k**) concentration overpotential under 15 °C, 25 °C, and 35 °C.

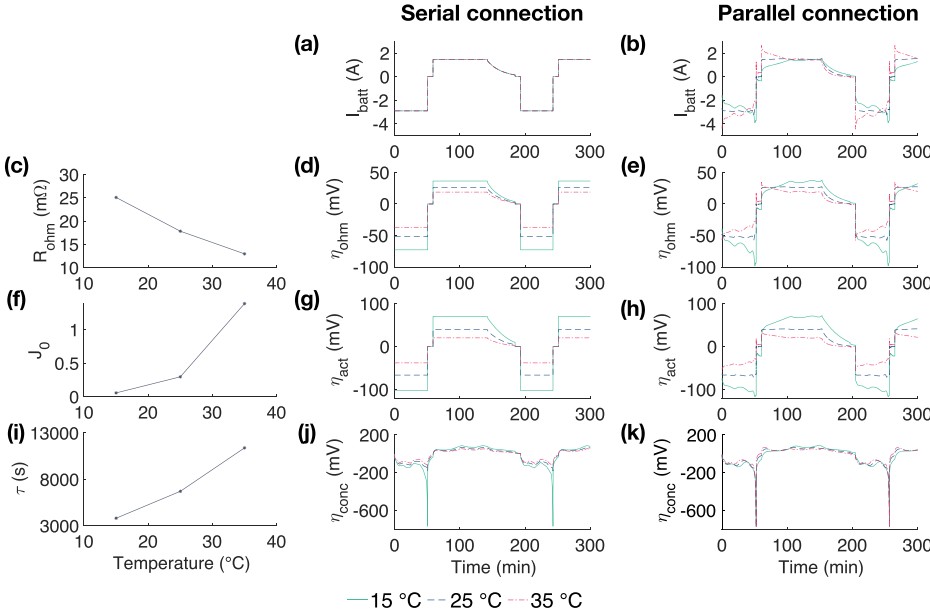

Figure 5i illustrates $\tau$ values of 3812 s, 6712 s, and 11,390 s at 10 °C, 25 °C, and 35 °C, respectively. This is attributable to the diffusion coefficient increasing with temperature, which leads to enhanced ion transport in the electrolyte[47]. In the LSPM, $\tau$ is defined as the inverse of the diffusion coefficient[38], therefore, it decreases as temperature rises. An increase in temperature by 20 °C results in an approximate change of 379 s °C$^{-1}$ in $\tau$. The $\eta_{conc}$ change during a 1C discharge at 10 °C, 25 °C, and 35 °C for serial and parallel connections is depicted in Fig. 5i, k, respectively.

**Influence of temperature gradient on ageing rate**

The current and voltage maldistribution due to temperature gradient, which leads to the inhomogeneous ageing, has been discussed in Section 'Influence of temperature gradient on electrochemical parameters'. Figure 6 demonstrates the influence of temperature ($f_T$), voltage ($f_E$), current ($f_I$), and ageing history ($f_{aged}$) on the serial and parallel connections, respectively. It also shows the resulting inhomogeneous cycling ageing rate at 15 °C, 25 °C, and 35 °C, following the same duty cycle as indicated in Section 'Influence of temperature gradient on electrochemical parameters'.

Figure 6a illustrates the ageing rate of $f_T$, showing that temperature exerts the same impact on the ageing rate for both configurations. It is noteworthy that $f_T$ increases at both the lower and higher temperature and is more noticeable at lower temperatures. Similarly, Fig. 6b illustrates the ageing rate of $f_{aged}$, which is caused by the side reactions, i.e. SEI growth, and demonstrates a comparable effect on the ageing of both serial and parallel connections. Notably, the most substantial reduction in the ageing rate is noted during the CV charge phase. This indicates that the cell experiences the most substantial capacity loss during this phase due to the increasing intercalation fraction of the negative electrode, leading to accelerated film growth during charging[48]. During discharge, the intercalation fraction decreases, slowing down the growth of the SEI film and leading to a more stable ageing rate of $f_{aged}$. Over a long-term scale, $f_{aged}$ decreases due to the SEI growth.

The underlying factors accounting for the divergent ageing behaviour between serial and parallel connections are the maldistribution of current and voltage. The detailed results of overpotential and voltage for each connection can be found in Supplementary Fig. S1 of the Supplementary Information. Eq. (20) indicates the current-related ageing solely depends on the applied current when the initial cell capacities are equivalent. As a result, current-related ageing (see Fig. 6c) exhibits a close alignment with the applied current. This synchronicity reveals that current distribution impacts on inhomogeneous cycling ageing in different connections.

On the voltage side, Eq. (18) indicates the ageing rate increases with $E_{OCV}(SOC_{ave}, T)$ and $\eta_{act}$, and decreases with temperature. $E_{OCV}(SOC_{ave}, T)$ is the primary parameter affecting the voltage-related ageing rate, which is the sum of $E_{OCV,ref}(SOC_{ave})$ and $E_{entropy}$ according to Eq. (10). However, the impact of $E_{entropy}$ tends to be marginal in light of their relatively small magnitudes. For parallel connection, voltage-related ageing rate increases with temperature due to $E_{OCV}(SOC_{ave})$ reaching its maximum value at 35 °C. In a serial connection, the variation of $E_{OCV}(SOC_{ave})$ primarily results from $E_{entropy}$; the influence of which is minimal and can typically be overlooked. Therefore, it's safe to assume that the open-circuit voltage of the cells ($E_{OCV}(SOC_{ave})$) is essentially identical for the serial-connected cell. Under such conditions, $\eta_{act}$ becomes the key factor impacting voltage-related ageing, which peaks at 15 °C, and in turn, leads to the highest ageing rate, as illustrated in Fig. 6d.

The overall ageing rate can be derived as the product of $f_T$, $f_E$, $f_I$, and $f_{aged}$, as illustrated in Fig. 6e. This allows for the derivation of the inhomogeneous cycling ageing in both serial and parallel connections under the fixed temperature. Figure 6f shows that SOH differences for parallel-connected cells are calculated as 0.03 between 15 °C and 25 °C, and 0.06 between 25 °C and 35 °C. In contrast, the SOH differences for serially connected cells are observed to be 0.05 for both temperature ranges, from 15 °C to 25 °C, and from 25 °C to 35 °C. For battery modules, however, the temperature difference among cells is not constant. To gain a deeper understanding of the impact of temperature gradients on the module-level inhomogeneous cycling ageing, the following section will investigate and discuss these effects under varying temperature gradients.

**Module-level current, voltage, and temperature distributions**

The coolant is applied to the module with an inlet temperature of 25 °C and a flow rate of 0.006 m s$^{-1}$. The duty cycle C-rate is same as the ageing cycling profile illustrated in Fig. 4c. Figure 7 illustrates the variations in temperature, current, voltage, and SOC during the 1C discharge for both the straight and parallelogram connection topologies. Figure 8 illustrates the temperature, current, voltage, and SOC distribution of the module.

Figure 7a illustrates that the cell temperature increases from 25 °C to a maximum of 34 °C by the end-of-discharge. Cell temperature increases as the cooling channel distance increases, with a maximum gradient of 5.04 °C at the end-of-discharge. The temperature difference between each cell is negligible, with a temperature variation of less than 0.62 °C between the parallelogram and straight connection topologies. Therefore, it shows that the temperature has a similar impact on the ageing rate for both connection

**Fig. 6 | Influence of temperature gradient on ageing rate (serial.** Comparison of the ageing factors of (**a**) $f_T$, (**b**) $f_{aged}$, (**c**) $f_I$, (**d**) $f_E$, and (**e**) f, and (**f**) the resulting inhomogeneous for serial connection and parallel connection under 15 °C, 25 °C, and 35 °C.

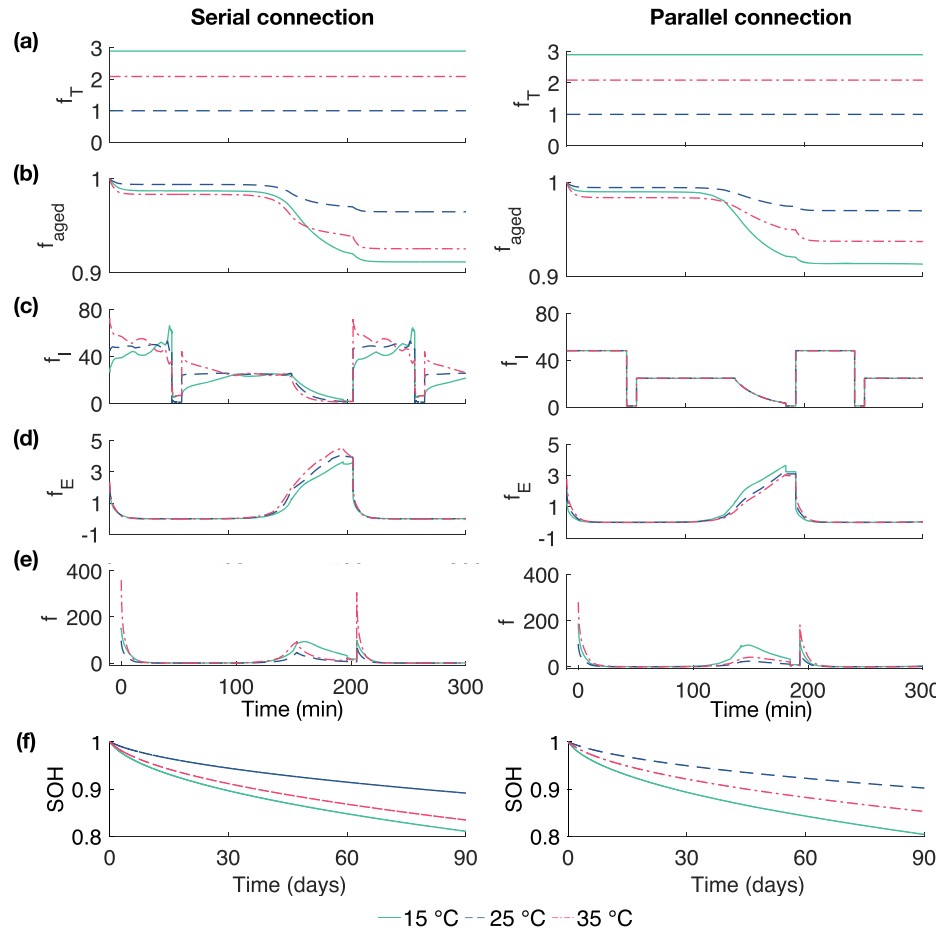

topologies at the cell level. The detailed temperature distribution of the module at 3214 s for the straight and parallelogram connection topologies is illustrated in Fig. 8a.

Figure 7b illustrates the current distribution among cells during the 1C discharge phase, where the negative sign (-) indicates discharge. It can be observed that cells on the side with a higher temperature experience an increase in $I_{batt}$ and reach their deep discharge state first (approximately 0.6 DOD). Beyond this threshold, the capacity of the cell operating at the higher temperature nears its full discharge state, leading to an increase in $I_{batt}$ for cells on the cooler side and a decrease for those on the hotter side. Figure 8b further illustrates the current distribution at 600 s and 3214 s to better depict this phenomenon. Detailed current distribution data for sub-module P1 can be found in Supplementary Fig. S2 of the Supplementary Information. Yang et al.[49] demonstrated similar results in their 2D model. When comparing different electrical connection topologies, the straight connection topology exhibits a more pronounced current maldistribution, with variations of up to 0.7 A (i.e. 0.24C) due to greater temperature gradients among sub-modules, while the parallelogram connection topology shows variations of less than 0.17 A (i.e. 0.05C). These distinct current distributions have varying impacts on current-related ageing, resulting in inhomogeneous cycling ageing.

It is worth noting that, to isolate the variables, interconnection resistance is neglected in this study. However, the proposed 3D model can also assess interconnection resistance by setting the connector's electrical conductivity to its original material value. Nevertheless, the study of interconnection resistance can be case-sensitive, as it depends on the selected material's electrical properties, geometry, and joining techniques[50]. This study primarily focuses on exploring representative electrical connection topologies, and the influence of interconnection resistance is beyond the scope of this research. To better demonstrate the influence of interconnection resistance, a sensitivity analysis is conducted by comparing

current distribution at 1C discharge using three different materials: steel, aluminium, and a hypothetical super-low resistance material, the latter of which is used in this study. The current distribution can be found in Supplementary Fig. S3 of the Supplementary Information. The results indicate that current maldistribution decreases with increasing electrical conductivity; however, the difference in current distribution between aluminium and the hypothetical super-low resistance material is negligible, justifying why we have opted to neglect interconnection resistance. This also underscores the reason why aluminium is one of the most used materials[51]. Thus, when selecting a material with high electrical conductivity (e.g. aluminium), interconnection resistance is not the primary factor influencing the inhomogeneous ageing between the straight and parallelogram connection topologies for this study.

Figure 7c illustrates the heat generation rate among cells during the 1C discharge phase. The heat generation rate of a cell depends on the current and temperature, as described in Eqs. (23)–(29). It is observed that cells on the cooler side exhibit a higher heat generation rate due to experiencing greater overpotential. Long-term, a cell's internal resistance increases during the ageing process. In Raj et al.'s study[52], a maximum increase of 10% was observed in the electrochemical impedance spectroscopy (EIS) test for two groups of NCA/graphite 18,650 cells, which were cycled at 0.25C and 0.5C, respectively, from an SOH of 1 to 0.83 under 24 °C. To investigate the influence of increased internal resistance on temperature distribution, a comparison between fresh cells and aged cells (with 10% increased internal resistance) is presented in Supplementary Fig. S4 of the Supplementary Information. The results show that the maximum temperature difference is 0.8 °C, indicating that the temperature distribution pattern is not significantly affected by the aging process.

Figure 7d illustrates the unbalanced SOC due to the different currents flowing through each cell. Despite the current on the cooler side being higher

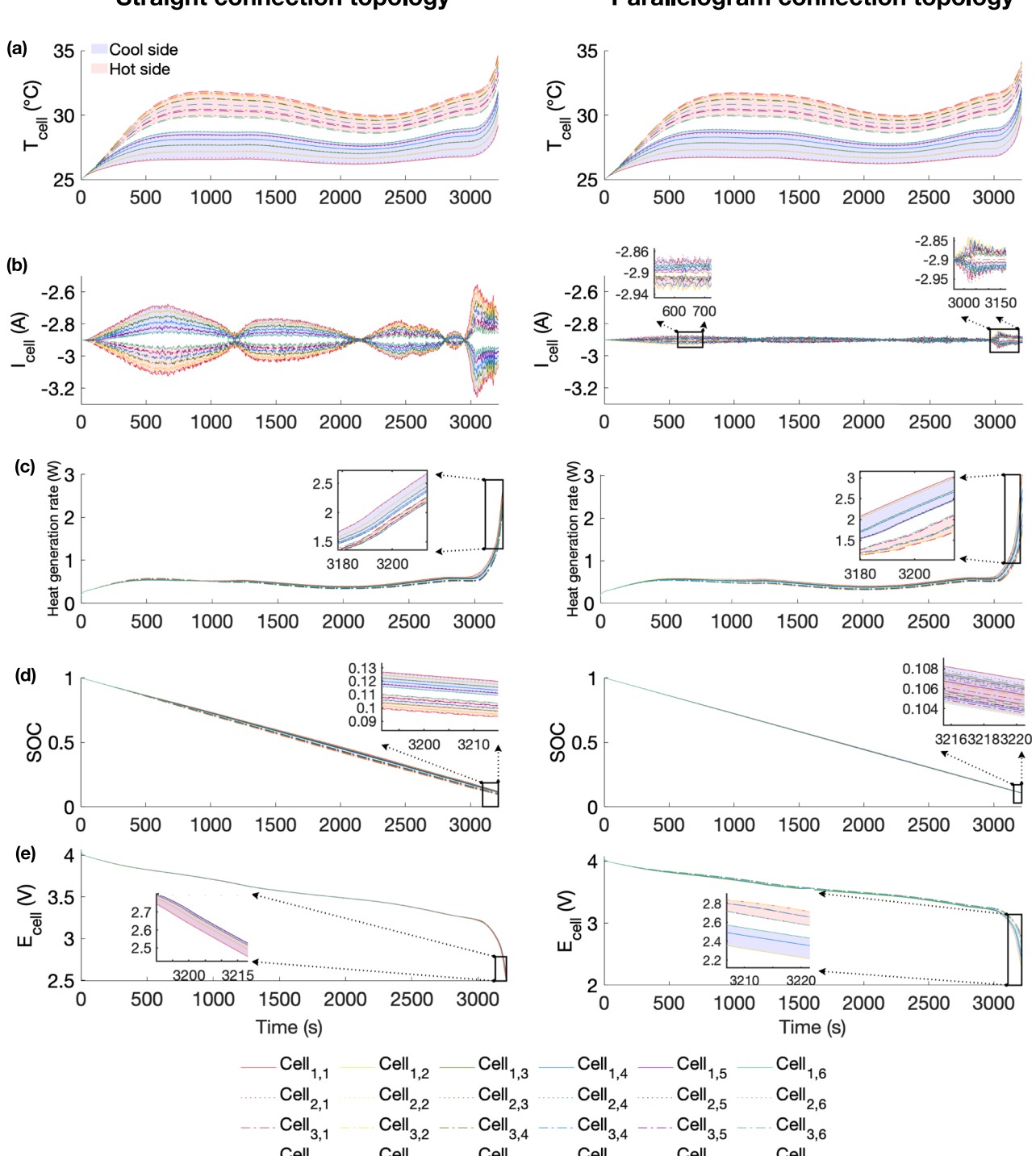

**Fig. 7 | Comparisons of module-level temperature, current, heat generation rate, state of charge (SOC), and voltage change for straight and parallelogram connection topologies under 1C discharge. a** Temperature increases from 25 °C to a maximum of 34 °C and temperature difference between two topologies is negligible (<0.62 °C). **b** Current change shows the straight connection topology exhibits higher maldistribution due to higher temperature gradient among cells within a sub-module due to higher temperature gradient among cells within a sub-module. **c** Heat generation rate change shows cells on the cool side have higher heat generation rate due to experiencing higher overpotential, and higher current maldistribution leads to higher heat generation rate gradient. **d** SOC change shows the straight connection topology exhibits a higher SOC gradient at the end-of-discharge due to current maldistribution. **e** Voltage change shows the parallelogram connection topology exhibits a higher voltage maldistribution due to higher temperature gradient among sub-modules.

after the DOD reaches 0.6, the cells on the hotter side consistently maintain a higher DOD throughout the entire discharging phase. In Fig. 8c, the straight connection topology exhibits a higher SOC gradient with a maximum SOC difference of 0.02 at the end-of-discharge, whereas the parallelogram connection topology results in a maximum SOC difference of 0.003.

Figure 7e illustrates the voltage changes during the discharging phase. The temperature gradient across these sub-modules results in an inhomogeneous voltage distribution within the battery module, causing the most heavily used cells to reach their safety voltage limits sooner. Figure 8d illustrates the voltage distribution for cells and electrical connectors at

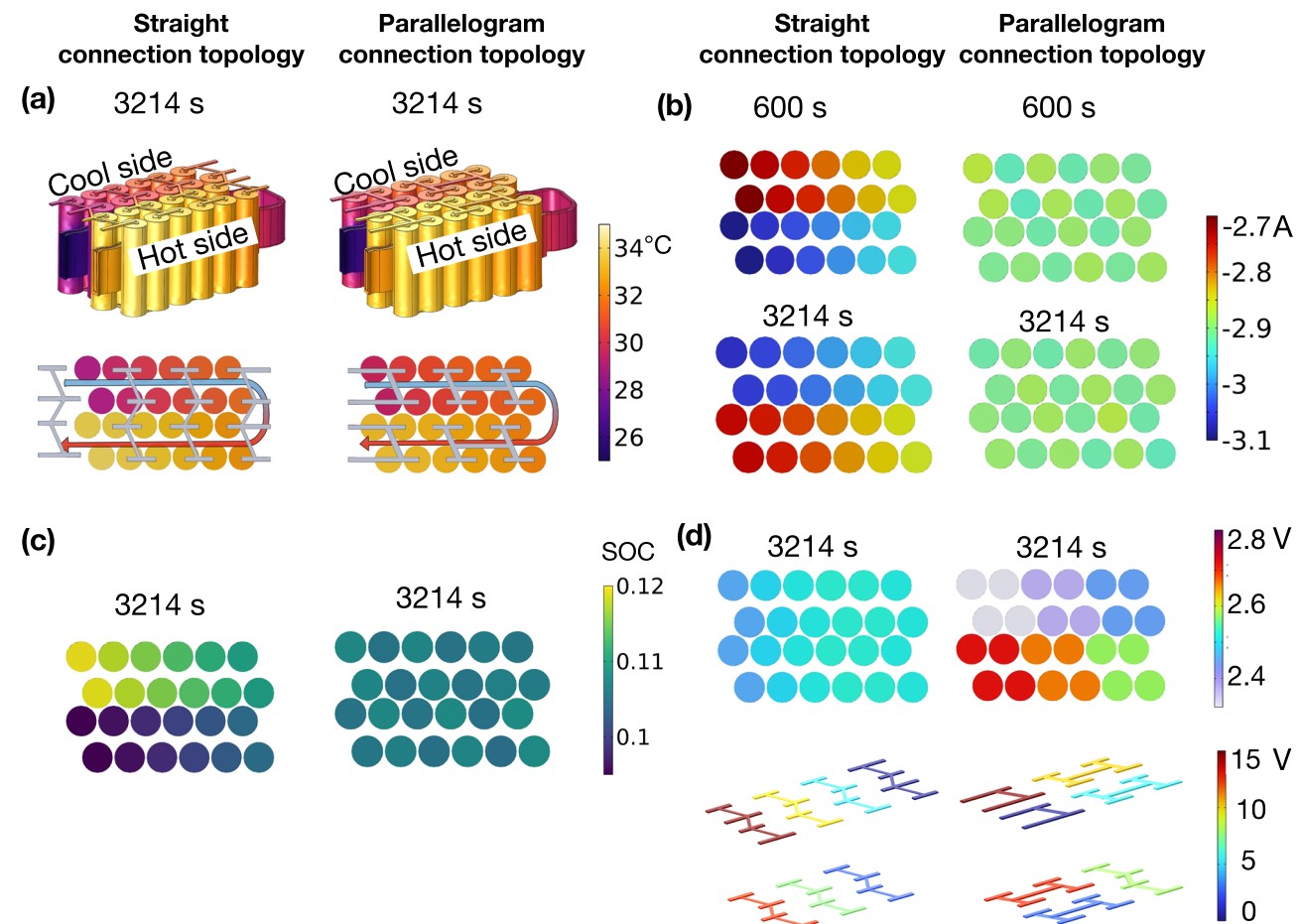

**Fig. 8 | The temperature, current, voltage, and state of charge (SOC) distribution of the module. a** Temperature distribution for the straight and parallelogram connection topologies at 3214 s. **b** Current distribution for the straight and parallelogram connection topologies at 600 s and 3214 s. **c** SOC distribution for the straight and parallelogram connection topologies at 3214 s. **d** Cell voltage and electrical connectors potential distribution for the straight and parallelogram connection topologies at 3214 s.

3214 s. It reveals that cells connected in parallel share the same terminal voltage. However, at the sub-module level, the parallelogram connection topology exhibits a more pronounced voltage difference of 0.138 V among its sub-modules, due to the temperature gradient among sub-modules being less than 1.21 °C, while the voltage difference among sub-modules is 0.016 V at the end-of-discharge for the straight connection topology due to a temperature gradient of less than 0.38 °C among the sub-modules.

As discussed in Section 'Influence of temperature gradient on electrochemical parameters', a cell's overpotential decreases with increasing temperature. Thus, sub-module P6 with the highest temperature exhibits the highest voltage in the parallelogram connection topology. It is also noteworthy that the lowest cell voltage is 2.22 V for the parallelogram connection topology and 2.45 V for the straight connection topology. The voltage of sub-module P1 falls below the minimum voltage limit (2.5 V) for both topologies, as the discharge endpoint depends on the module voltage value. This indicates that temperature gradients can potentially lead to over(dis)charging of cells at the lowest temperatures, resulting in over(dis)charging at the sub-module level due to the parallel connection of cells. This highlights the necessity of controlling the minimum and maximum module voltages to prevent cell abuse, especially in designs with high-temperature gradients among the sub-modules (i.e. the parallelogram connection topology in this study). However, this precaution would reduce the effective capacity of the battery module. In this study, if the discharge stops when the first cell reaches the cut-off voltage, the parallelogram connection topology has an effective discharge capacity of 88.6%, while the straight connection topology has an effective discharge capacity of 89.4% at a 1C discharge,

resulting in a 0.8% capacity difference. These findings agree with Marlow et al.'s study[12] that within parallel-connected cells, accessible capacity is reduced due to the end-of-discharge SOC deficit.

**Module-level inhomogeneous cycling ageing**

As discussed in Section 'Module-level current, voltage, and temperature distributions', the straight connection topology exhibits a higher current maldistribution among parallel-connected cells, but it demonstrates a more even voltage distribution among serial-connected sub-modules. The parallelogram connection topology ensures a more even current distribution among parallel-connected cells, but it shows a higher voltage maldistribution. Figure 9a illustrates that the overall SOH distribution for the straight and parallelogram connection topologies is similar due to them experiencing the same temperature distribution (see Fig. 8a). Figure 9b provides a detailed comparison of the overall ageing trend for both topologies. The maximum inhomogeneous cycling ageing difference of 1% is observed for both topologies, with a maximum temperature gradient of 4.96 °C. However, the parallelogram connection topology exhibits a higher ageing rate (0.1% SOH) than the straight connection topology, resulting in a premature end-of-life.

For the straight connection topology (see Fig. 9c), the side with cooler temperatures displays a more pronounced inhomogeneous cycling ageing spread (Δ SOH = 0.5%) compared to the side with higher temperatures (Δ SOH = 0.2%). This disparity is primarily attributed to the higher current maldistribution (see Fig. 9b), resulting from a higher temperature gradient within the sub-modules (see Fig. 8a). In contrast, the parallelogram

**Fig. 9 | The state of health (SOH) spread for the straight connection topology and parallelogram connection topology. a** The 3D SOH distributions for each topology when the first cell reaches the end-of-life. **b** The parallelogram connection topology exhibits a higher ageing rate than the straight connection topology (0.1% SOH). **c** The straight connection topology exhibits a more pronounced spread of inhomogeneous cycling ageing on the cool side (Δ SOH = 0.5%) than the hot side (Δ SOH = 0.1%) due to the current maldistribution. **d** The parallelogram connection topology exhibits a more uniform spread of inhomogeneous cycling ageing on the cool side (Δ SOH = 0.5%) and the hot side (Δ SOH = 0.4%) due to a more uniform current distribution resulting from a more uniform temperature distribution within the sub-modules.

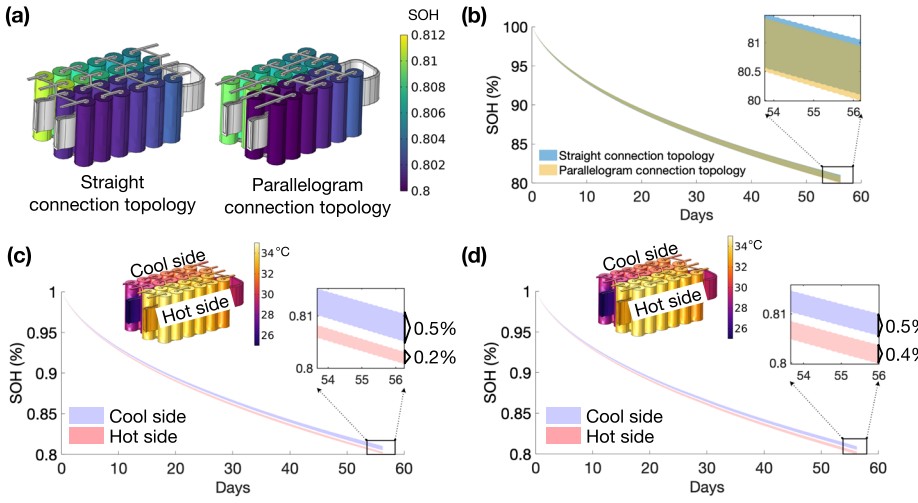

connection topology (Fig. 9d) exhibits a more balanced inhomogeneous cycling ageing spread on both sides due to a more uniform current distribution. The inhomogeneous cycling ageing difference for the cool and hot sides are 0.5% and 0.4%, respectively.

It worth pointing out this study focuses on the cycling ageing with a CC-CV profile due to cells experiencing same calendar ageing rate in their resting condition. The ageing model only considers capacity loss due to SEI growth as it is the main ageing factor in most graphite-based lithium-ion batteries. Lithium plating is not considered, as it mainly occurs under high C-rate or low-temperature conditions, where the C-rate is under 1C, and the temperature is above 25 °C in this study. Calendar ageing rate is not considered in this study as cells experience same ageing rate when the temperature distribution is homogeneous in their resting condition at the same SOC.

## Conclusions

This study adopts experimental CC-CV cycling ageing data and numerically investigates the effects of different electrical connection topologies on inhomogeneous cycling ageing caused by thermal gradients in 2D models configured in 1P3S and 3P1S, and a 3D model configured in 4P6S. To isolate variables, cell-to-cell variances caused by intrinsic factors (i.e. capacity, internal resistance) and extrinsic factors (i.e. interconnection resistance and welding resistance) are not considered. Our results show that while temperature gradients remain consistent for a given BTMS and are the primary factor driving inhomogeneous cycling ageing, important variations in the inhomogeneous cycling ageing rate under the same temperature gradient stem from differences in electrical connection topology. Specifically, the 2D model emphasises that current maldistribution is the main factor behind inhomogeneous cycling ageing in parallel-connected cells, whereas voltage maldistribution primarily affects serially-connected cells.

The 3D model highlights the fact that the straight connection topology maintains a more uniform voltage distribution among serially-connected sub-modules, which is due to temperature gradients being more pronounced among individual cells within a sub-module. However, this leads to a pronounced current maldistribution among parallel-connected cells. Consequently, a wider spread of inhomogeneous cycling ageing is exhibited on the cooler side and a more concentrated spread of inhomogeneous cycling ageing on the hotter side compared to the parallelogram connection topology. In contrast, the parallelogram topology ensures a more uniform current distribution among parallel-connected cells but exhibits higher voltage maldistribution due to temperature gradients being more pronounced between sub-modules. A notable limitation of the parallelogram topology is the over(dis)charging issue for the sub-module with the lowest temperature due to it experiencing the highest voltage drop. Thus, it is

essential to narrow the minimum and maximum module voltages, which ultimately reduces the battery module effective capacity. Consequently, a 0.8% reduced effective capacity is found for the parallelogram topology compared to the straight connection topology at a 1C discharge with a max temperature gradient of 4.96 °C.

On balance, we find that the straight electrical connection topology is likely to be advantageous. Most importantly, by allowing the temperature gradient to be distributed among the parallel-connected cells in the sub-modules, it mitigates the over(dis)charge issue, resulting in a higher effective capacity (e.g. an increase of 0.8% in capacity compared to the parallelogram connection topology at 1C discharge). It also exhibits a higher SOH of 80.15% compared to 80% for the parallelogram connection topology under the same CC-CV cycling profile. It is important to note that the straight topology does result in increased current maldistribution within sub-modules (i.e. a maximum current maldistribution of 0.24C at 1C discharge compared to a maximum of 0.05C for the parallelogram connection topology), but it is considered an acceptable trade-off. The future work will focus on module-level experimental validation to further examine the current maldistribution and inhomogeneous ageing within the proposed battery module.

## Data availability

All processed data generated as part of this work is available and can be found at https://doi.org/10.5281/zenodo.10935848. Detailed descriptions of the COMSOL models developed for this work are provided in 'The guideline.pdf' within the repository. To reduce the file size, the results within the COMSOL model may be cleared and will need to be rerun on COMSOL 6.2 or above to obtain the results again. All raw data is available in a. txt format and processed data can be found in. mat format.

## Code availability

All codes used in processing raw data and generating figures as part of this work is available and can be found in 'matlab code.zip' at https://doi.org/10.5281/zenodo.10935848. Data processing and visualisation were completing on MATLAB r2023b.

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

## Acknowledgements
This work acknowledges the use of the Lovelace HPC service at Loughborough University.

## Author contributions
H.H.: data analysis, battery modelling, parameter estimation, visualisation, writing - Original Draft, writing -review & editing, A.F.: conceptualisation, methodology, supervision, writing - review & editing. E.B.: writing - review & editing. X.C.: supervision, writing -review & editing.

## Competing interests
The authors declare no competing interests.
