## [Peer Review File · Communications Engineering]

Reviewers' comments:

Reviewer #1 (Remarks to the Author):

This paper presents a model-based investigation on the impact of electrical connection topology on inhomogeneous ageing under CCCV cycling conditions. Although the authors just present a study case, the topic is interesting, since most of the studies in the literature focus on cell-level issues, and therefore module – and pack-level design issues are relatively unexplored.

The methods are fine in general, but there are some major issues to be reviewed, including certain assumptions and the thermal model parameterization, which is poorly done, as I detailed in followings.

In my view, the title should be made more specific, and the abstract and conclusions should be re-written, in order to highlight the key limitations of the study, i.e.:

- 1) it is a model-based and not an experimental study,
- 2) it is not a universal study, but a study case, providing just incremental insights,
- 3) focuses on cycling ageing with a CCCV profile, and disregards calendar ageing,
- 4) resistance interconnections are not considered, and
- 5) the ageing model only considers capacity loss based on SEI layer growth.

The geometric dimensions of the modules are not provided, and this means that the results cannot be reproduced without access to the original simulation files.

My recommendation is to accept with major changes.

Comments to Introduction. Pages 2-4.

1. “Notably, higher C-rate (i.e. $> 2C$), high state of charge (SOC) (i.e. > 0.75 SOC), and extreme temperatures (i.e. $> 40\text{ }^{\circ}\text{C}$ or $< 10\text{ }^{\circ}\text{C}$) accelerate the ageing rate [1, 3]. (...) Therefore, BTMSs need to be carefully designed to maintain optimal cell temperature ranges (i.e. $20\text{ }^{\circ}\text{C} - 40\text{ }^{\circ}\text{C}$ [5]) and ensure optimal performance and longevity of the batteries.”

This is oversimplified. The interactions between current, SOC and temperature influence calendar and cycling ageing differently. For example, low temperatures decelerate calendar ageing. This also influences the optimal operating temperature, which is actually variable within a wide range of temperatures, as low as $10\text{ }^{\circ}\text{C}$ in the case of an application with low C-rates.

This issue is briefly commented later in methods, but in my view should be discussed here: “The temperature-related ageing rate varies under cycling or resting conditions and is higher at lower temperatures during cycling [28], but lower at lower temperatures during calendar ageing [29]. This study focuses exclusively on cycling ageing.” (Page 4, third paragraph).

2. “To provide useable power outputs for EVs, cells are parallel/series-connected at module level, which

can significantly reduce the cycling C-rate while meeting the same power requirements [4].”

For the same number of cells, the C-rate is the same if the cells independent of the series-parallel arrangement.

3. “Liquid cooling, as a prevalent BTMS, provides high cooling efficiency, improved maintainability, moderate power usage, quick thermal response, and adaptability to both cooling and preheating conditions [6, 7].”

I assume that in comparison with forced air cooling, but the authors should clarify. There are different ways to implement liquid cooling, and also other means, e.g. immersion cooling. However, the later is not mentioned, and the former very briefly and not explicitly. I would recommend elaborating a bit more on that. Suggested to take a look to C. Roe et al. [R3] for ideas.

4. “For the temperature distribution regulation, previous research has primarily focused on optimisation strategies of cell layouts [17, 18], flow patterns [19, 20], and BTMS geometries [21, 22].”

I would say that there has been a lot of work on other topics, such as battery thermal modelling and parameterization (thermal conductivity and heat capacity), cell temperature inhomogeneities, preheating and self-heating methods, cell tab design, interconnection resistances, battery mechanical design, battery re-configuration methods, or novel balancing methods that aim to control temperature.

5. “These aforementioned studies highlight that, under the same cooling conditions, disparities in ageing rates can be further managed by modifying the electrical configurations. (...) Therefore, it is crucial to build a 3D battery module model with both parallel and series connections while considering real-time temperature changes to gain a comprehensive understanding of the module-level uneven ageing.”

Well, based on all the previous studies, we know that many factors influence the cell-to-cell differences in terms of ageing, and all of them should be taken into account, i.e. engineers should aim for a holistic design approach to tackle this problem. The authors should clarify that this paper focuses in one aspect and one study case only.

6. “Leading EV manufacturers, Lucid and Tesla, employ...”

I would delete the word “leading”.

Comments to Methods section. Pages 4-10.

7. “Before the numerical simulation, the following assumptions have been made in this study as follows: 1. (...) 2. Cell ohmic resistance changes due to ageing is not considered, given that only a 5% increase was observed when the SOH decreased from 1 to 0.8 in Teliz et al.’s study [30].”

The validity of this assumption is not clear for me, since the data used for model parameterization is apparently coming from other sources [32, 36, 37] and based on a specific Panasonic cell, while in [30]

no cell is specified.

8. “Before the numerical simulation, the following assumptions have been made in this study as follows:
1. (...) 3. Interconnection resistance is not considered in this study due to the focus on investigating the impact of electrical topologies on inhomogeneous ageing. In the 2D model, the connector resistance is neglected. In the 3D model, the electrical conductivity of connectors is manually set to an extremely high value of $1 \times 10^{12} \text{ Sm}^{-1}$ to minimise the voltage drop and ohmic heating across the connectors.”

The validity of this assumption is not clear for me, since in the series-parallel arrangement in the two packs considered is not the same, and therefore the interconnection resistances could play a role on inhomogeneous ageing, as shown in previous literature, e.g. in [16].

9. “Before the numerical simulation, the following assumptions have been made in this study as follows:
1. (...) 4. Heat conduction is the only heat transfer mode considered in this simulation.”

To ease understanding, the authors should clarify here if this means if what are the actual paths considered for heat transfer by conduction, e.g.: 1) cell-to-cell through the surface, 2) cell-to-cell through the busbar, 3) cell-to-cooling system, etc. I know the answer after reading the whole paper, but the authors should make clear here if they are assuming: 1) a lumped thermal cell model, i.e., if they use a volume averaged temperature, or a 2) 3D distributed thermal cell model, and reflect somewhere on what are the consequences of that in the results obtained.

A diagram explaining how the 2D and the 3D model interact would be useful too, to ease understanding.

10. Regarding other possible assumptions to be listed.

- Cell-to-cell differences are not considered, although they are well documented in the literature, e.g. in [R4]. This assumption could be fine, but this should be included in the list and the consequences of that in the results commented clearly.
- The ageing model only considers SEI growth, which might be fine too, but it should be highlighted and listed.

11. Regarding the thermal model parameterization: heat capacity and thermal conductivities. Considering the type of model-based study that the authors are presenting, careful parameterization of the thermal model is mandatory.

The authors justify the cell heat capacity parameter value of 750 J/kg-K based on [46], but: 1) in that reference a lower value is provided (727 J/kg-K), 2) this is not an appropriate reference, since it is an investigation based on a fake/surrogate cell, and not a real cell, and 3) in [46] it is mentioned a wide range of possible values for heat capacity found in the literature $800\text{-}1700 \text{ J/kg-K}$. Authors are encouraged to find a better reference to justify the election and change the value if needed.

Regarding the axial/radial thermal conductivities, the authors justify the selection of the parameters based on reference [47], i.e. M. Al-Zareer “Numerical Study of Flow and Heat Transfer Performance of

3D-Printed Polymer-Based Battery Thermal Management.” Based on that thereference too, the heat capacity of the cell is 1400 J/(kg · K), which is contradictory with previous assumptions. Moreover, in this reference, the author justifies the values of the heat capacity and thermal conductivities based on references:

- M. Doyle, T.F. Fuller, J. Newman, “Modeling of Galvanostatic Charge and Discharge of the Lithium/Polymer/Insertion Cell,” J. Electrochem. Soc., 140 (6) (1993), p. 1526
- M. Doyle, “Comparison of Modeling Predictions with Experimental Data from Plastic Lithium Ion Cells,” J. Electrochem. Soc., 143 (6) (1996), pp. 1890-1903.
- T.F. Fuller, M. Doyle, J. Newman, “Simulation and Optimization of the Dual Lithium Ion Insertion Cell”, J. Electrochem. Soc., 141 (1) (1994), pp. 1-9.

However, in none of these references there is data about thermal parameters.

The authors are suggested to take a look to [R5] to find references to justify their selection of parameters.

Changing the values of these parameters necessarily means that the authors will have to repeat the simulations and re-write the results section. The current parameters are not valid.

Comments to results (section 2.4 Module-level inhomogeneous ageing with real-time temperature changes). Pages 16-19.

The authors have implemented a 3D model in COMSOL, but the authors do not present any figure in the paper that shows actual 3D results of the temperature distribution with inhomogeneous cell temperatures. I would suggest including that, and not only 2D results plots (Fig. 7 and 8).

Comments to section 3. Conclusion. Page 20.

The key limitations and assumptions, and their impact in the results, should be commented in the the conclusions - or a discussion section could be included in the paper.

REVIEWER REFERENCES

[R1] Charlotte Roe, Xuning Feng, Gavin White, Ruihe Li, Huaibin Wang, Xinyu Rui, Cheng Li, Feng Zhang, Volker Null, Michael Parkes, Yatish Patel, Yan Wang, Hewu Wang, Minggao Ouyang, Gregory Offer, Billy Wu, Immersion cooling for lithium-ion batteries – A review, Journal of Power Sources, Volume 525, 2022, 231094, ISSN 0378-7753, <https://doi.org/10.1016/j.jpowsour.2022.231094>.

[R2] M. Schindler, J. Sturm, S. Ludwig, J. Schmitt, A. Jossen, Evolution of initial cell-to-cell variations during a three-year production cycle, eTransportation, Volume 8, 2021, 100102, ISSN 2590-1168, <https://doi.org/10.1016/j.etrans.2020.100102>.

[R3] Marco Steinhardt, Jorge V. Barreras, Haijun Ruan, Billy Wu, Gregory J. Offer, Andreas Jossen, Meta-analysis of experimental results for heat capacity and thermal conductivity in lithium-ion batteries: A critical review, *Journal of Power Sources*, Volume 522, 2022, 230829, ISSN 0378-7753, <https://doi.org/10.1016/j.jpowsour.2021.230829>.

Reviewer #2 (Remarks to the Author):

The authors have developed 2D and 3D models to investigate the impact of serial and parallelogram design architecture on inhomogeneous aging at the module level. The physical baseline of the models is well explained; however, it is unclear what are the novel contributions of the work. The overall quality of the writing and manuscript structure is good. Further comments are listed below for the reviewed work.

- The abstract fails to introduce the straight and parallelogram design so that the results make sense. The authors should also rewrite the last two lines, especially because 'due to limiting the voltage range' is unclear.
- The authors are recommended to focus on BTMS only to provide the baseline study reasoning. At times, the discussion seems unnecessarily engaged in the thermal gradient, BTMS, etc. Otherwise, the title of the work should be rephrased.
- The topology type should be in a consistent format like $\alpha P\beta S$; not 15p1s (and similar on page 3).
- In section 1, point 1 – the authors assume a literature-driven aging rate; however, this is cell-specific. Same comment for the point 2. Are these studies based on the same cell? How to justify the assumptions when the research topic is about inhomogeneous aging on the cell level?
- Table A1 is mentioned to be taken from ref [32] in the text but the table itself refers to [45]. The authors should use only one reference.
- The bottom views of Fig. 3d and 3e have no significance, thus, suggested to be removed.
- Typo in the text on page 11 and in Fig. 4c where the cold temperature is said to be 15°C, and 10°C. Only one value should be correct, this, unfortunately, creates confusion.
- The theoretical models' generated influential factor calculation is made and the numbers are mentioned in the text but no physical explanations are available for many cases. The authors are encouraged to dive deeper into the aging mechanism.
- The model-based results are not validated in physical measurement. This is a big shortcoming. The data from the literature must be verified with the specified test for the specified topologies.
- The novel contribution of the research seems also to require bigger motivation as most of the works are taken from different literature.
- The authors should also comment on additional module-level factors like cell-to-cell variation, pressure, electrical connection, vibration, etc. which are to be discussed as possible reasons behind inhomogeneous aging.

Reviewer #3 (Remarks to the Author):

The authors investigate the unbalanced distributions among the cells with two study cases, the straight

and parallelogram. The authors have done a very nice work and presented their results in a clear way.

Introduction main focus is the inhomogeneities from thermal management point of view, with the various cooling methods (air-liquid, which is also not SoA). More efforts should be placed on the effect of electrical connections/ bus bars/ materials/ soldering methods etc to align with the paper title. This is also contradicting with page 3 ("this study incorporates...under changing temperatures.)

Assumption 2 and 3: elaborate more on this. Why such important parameters are not considered? What are the limitations of considering only the capacity fade during ageing (and only by SEI) and how relevant is this capacity fade to electrical topology and nonuniform heat distribution? Which other ageing mechanisms the uniformities can affect?

Include a table with the parameters used for the electrical, thermal and ageing models and indicate their values and how they were obtained. This will increase the validity and reproducibility of the paper.

The thermal model is not clear and it needs improvements. How the surface temperature of the cells is derived? How did the thermal parameters are obtained? What is the reference temperature stated in the paper?

3D thermal model appears more as a TMS study instead of electrical connections/topology. What is the difference in the physics between the two models (straight and parallel)? How these two designs affect the heat generation of the cells or the conduction? More focus and explanations should be given on the electrical connections of the models. Do all the cells have the same parameters and states in both modules, i.e is a single cell extrapolated to the whole module?

Is the 3D module model is validated? What is the limitations here?

Fig. 5c and 5f. Why the Vohmic increases for the 35degC, while it decreases for the 15degC? Please indicate the charge-discharge profile with respect to time.

Section2.4, what is the natural convention (no cooling temperature behavior) of the cells/modules for both cases?

How is the balancing of the BMS could affect the conclusions of this paper? What is the cost of each module with respect to the components used?

At the particle level, Reviewer 1

General comments:

In my view, the title should be made more specific, and the abstract and conclusions should be re-written, in order to highlight the key limitations of the study, i.e.:

- 1) it is a model-based and not an experimental study,
- 2) it is not a universal study, but a study case, providing just incremental insights,
- 3) focuses on cycling ageing with a CCCV profile, and disregards calendar ageing,
- 4) resistance interconnections are not considered, and
- 5) the ageing model only considers capacity loss based on SEI layer growth.

Response: we have re-written the abstract and conclusions to reflect these changes, and highlighted in red Page 1 and 23-24. Main changes include:

- 1) *We have added “numerically investigated...” in Abstract and Conclusion to reflect this. The title of the paper has also been changed to highlight that this is a numerical study.*
- 2) *This study demonstrates the interaction between electrical connections, thermal management and ageing through the use of a representative test case, but is not a universal study. To better indicate this, we have modified the conclusions to state:*

“This study adopts experimental CC-CV cycling ageing data and numerically investigate the effects of different electrical connection topologies on inhomogeneous cycling ageing caused by thermal gradients in 1P3S and 3P1S 2D models, and a 4P6S 3D model, respectively. To isolate variables, cell-to-cell variances caused by intrinsic factors (i.e. capacity, internal resistance) and extrinsic factors (i.e. interconnection resistance and welding resistance) are not considered.”

- 3) *This paper focusses on inhomogeneous ageing under cycling because this is when the majority of inhomogeneities occur at the module/pack level. During steady state calendar ageing, there is minimal thermal gradient between cells, no current maldistribution and cell voltages are ideally balanced by the battery management system. We have modified the manuscript to better indicate this and to highlight that SOH refers to cyclic only:*

- *We changed 'inhomogeneous ageing' to 'inhomogeneous cycling ageing' in the manuscript.*
- *We updated our assumption point 3 and 4 (highlighted in red on Page 8):*

“3. The temperature-related ageing rate varies under cycling or resting conditions, and increases at lower temperatures during cycling [41], but decreases at lower temperatures during calendar ageing [42]. This study exclusively focuses on cyclic ageing due to cells experiencing the same calendar ageing rate when the temperature distribution is homogeneous and the cell is resting.

4. The ageing mechanism considered in this study is the SEI formation, which is the main ageing process in most graphite-based lithium-ion batteries [43]. Lithium plating is not considered due to it mainly occurring in low temperature or high C-rate conditions [43].”

Additionally, in the Conclusion, we emphasise that the ageing profile is based on 'CC-CV' ageing results.

4) The resistance of interconnections were not considered to present an idealised case that isolates the influence of topology design only and not topology and resistance combined, we have modified the manuscript to better highlight this point in the following ways:

We have added a discussion of interconnection resistance on Page 19:

“It is worth noting that, to isolate the variables, interconnection resistance is neglected in this study. However, the proposed 3D model can also assess interconnection resistance by setting the connector's electrical conductivity to its original material value. Nevertheless, the study of interconnection resistance can be case-sensitive, as it depends on the selected material's electrical properties, geometry, and joining techniques [62]. This study primarily focuses on exploring representative electrical connection topologies, and the influence of interconnection resistance is beyond the scope of this research. However, to better demonstrate the influence of interconnection resistance in this study, a sensitivity analysis is conducted by comparing current distribution at 1C discharge using three different materials: steel, aluminium, and a hypothetical super-low resistance material (i.e. in this study). The current distribution can be found in Fig. S3 of the Supplementary Information. The results indicate that current maldistribution decreases with increasing electrical conductivity; however, the difference in current distribution between aluminium and the hypothetical super-low resistance material is negligible, justifying why we have opted to neglect interconnection resistance. This also underscores the reason why aluminium is one of the most used materials [63]. Thus, when selecting a material with high electrical conductivity (e.g. aluminium), interconnection resistance is not the primary factor influencing the inhomogeneous ageing between the straight and parallelogram connection topologies for this study.”

Note: Fig. S3 can be found at the end of this document.

In Conclusion, we added:

“To isolate variables, cell-to-cell variances caused by intrinsic factors (i.e. capacity, internal resistance) and extrinsic factors (i.e. interconnection resistance and welding resistance) are not considered.”

5) To reflect ageing, we updated our assumption point 4 (highlighted in red on Page 8):

The ageing mechanism considered in this study is the SEI formation, which is the main ageing process in most graphite-based lithium-ion batteries [43]. Lithium plating is not considered due to it mainly occurring in low temperature or high C-rate conditions [43].

The geometric dimensions of the modules are not provided, and this means that the results cannot be reproduced without access to the original simulation files.

Response: We have updated Fig. 3 (e-f) with geometric dimensions on Page 7:

Figure 3 The 3D geometry of: (e) straight connection topology, and (f) parallelogram connection topology.

Bullet comments:

1. “Notably, higher C-rate (i.e. $> 2C$), high state of charge (SOC) (i.e. > 0.75 SOC), and extreme temperatures (i.e. $> 40\text{ }^{\circ}\text{C}$ or $< 10\text{ }^{\circ}\text{C}$) accelerate the ageing rate [1, 3]. (...) Therefore, BTMSs need to be carefully designed to maintain optimal cell temperature ranges (i.e. $20\text{ }^{\circ}\text{C} - 40\text{ }^{\circ}\text{C}$ [5]) and ensure optimal performance and longevity of the batteries.”

This is oversimplified. The interactions between current, SOC and temperature influence calendar and cycling ageing differently. For example, low temperatures decelerate calendar ageing. This also influences the optimal operating temperature, which is actually variable within a wide range of temperatures, as low as 10°C in the case of an application with low Crates.

This issue is briefly commented later in methods, but in my view should be discussed here: “The temperature-related ageing rate varies under cycling or resting conditions and is higher at lower temperatures during cycling [28], but lower at lower temperatures during calendar ageing [29]. This study focuses exclusively on cycling ageing.” (Page 4, third paragraph).

Response: We have rewritten the ageing review and discussed the ageing mechanism under cycling and calendar ageing, respectively. The revised content is highlighted in red on Page 2:

“Cell ageing can be categorised into two types: calendar ageing and cycling ageing. Calendar ageing occurs predominantly when the battery is not being used, while cycling ageing happens during charge or discharge, for example, when an EV is being charged or driven [1]. The calendar ageing rate increases with a higher state of charge (SOC) and temperatures, while it decreases over time due to the formation of the solid electrolyte interphase (SEI) layer, which follows the inverse of the square root of time dependence [2]. Conversely, the rate of cycling ageing is influenced by several factors: it increases with the C-rate, SOC, and higher temperatures, and also accelerates under lower temperatures due to lithium plating [2,3].

We have also updated ageing assumption in Point 4, and highlighted in red on Page 8:

“The temperature-related ageing rate varies under cycling or resting conditions, and increases at lower temperatures during cycling [41], but decreases at lower temperatures during calendar ageing [42]. This study exclusively focuses on cyclic ageing due to cells experiencing the same calendar ageing rate when the temperature distribution is homogeneous and the cell is resting.”

2. “To provide useable power outputs for EVs, cells are parallel/series-connected at module level, which can significantly reduce the cycling C-rate while meeting the same power requirements [4].”

For the same number of cells, the C-rate is the same if the cells independent of the series parallel arrangement.

Response: Thank you for highlighting this. We have removed this incorrect statement.

3. “Liquid cooling, as a prevalent BTMS, provides high cooling efficiency, improved maintainability, moderate power usage, quick thermal response, and adaptability to both cooling and preheating conditions [6, 7].”

I assume that in comparison with forced air cooling, but the authors should clarify. There are different ways to implement liquid cooling, and also other means, e.g. immersion cooling. However, the later is not mentioned, and the former very briefly and not explicitly. I would recommend elaborating a bit more on that. Suggested to take a look to C. Roe et al. [R3] for ideas.

Response: We have rewritten the BTMS review by comparing indirect liquid cooling to forced air cooling and direct cooling, respectively. The revised content is highlighted in red on Page 2:

“An effective battery thermal management system (BTMS) is crucial in EVs to maintain uniform temperature distribution within the battery pack. Indirect liquid cooling has become the most prevalent BTMS in commercial EVs to date [7, 8]. This method offers a quicker thermal response, higher cooling efficiency, and adaptability to both cooling and preheating conditions compared to forced air cooling [8, 9]. When compared to direct cooling (or immersion cooling), indirect cooling is favoured due to its easier implementation [10]. Moreover, the liquid coolant (e.g. ethylene glycol/water) has a lower viscosity than dielectric liquids (e.g. mineral oil), allowing for a much higher flow rate with fixed pumping power [10, 11].”

4. “For the temperature distribution regulation, previous research has primarily focused on optimisation strategies of cell layouts [17, 18], flow patterns [19, 20], and BTMS geometries [21, 22].”

I would say that there has been a lot of work on other topics, such as battery thermal modelling and parameterization (thermal conductivity and heat capacity), cell temperature inhomogeneities, preheating and self-heating methods, cell tab design, interconnection resistances, battery mechanical design, battery re-configuration methods, or novel balancing methods that aim to control temperature.

Response: We have added more literature to enrich this topic. The revised content is highlighted in red on Page 3:

“Various approaches have been proposed to regulate temperature homogeneity within the module, such as cell tab optimisation [31–33], low-temperature preheating/self-heating techniques [34, 35], battery mechanical design [36], reconfigurable battery management systems [37, 38], and novel balancing methods [6, 39]. From the perspective of the BTMS, various aspects such as cell layouts [40, 41], flow patterns [42, 43], and BTMS geometries [44, 45] have been studied as means to improve homogeneity.”

6. “Leading EV manufacturers, Lucid and Tesla, employ...”
I would delete the word “leading”.

Response: we have deleted “leading”.

7. “Before the numerical simulation, the following assumptions have been made in this study as follows: 1. (...) 2. Cell ohmic resistance changes due to ageing is not considered, given that only a 5% increase was observed when the SOH decreased from 1 to 0.8 in Teliz et al.’s study [30].”

The validity of this assumption is not clear for me, since the data used for model parameterization is apparently coming from other sources [32, 36, 37] and based on a specific Panasonic cell, while in [30] no cell is specified.

Response: We have updated a new literature from Raj et al. [52]. They measured the resistance change in NCA/graphite 18650 Panasonic cells using an electrochemical impedance spectroscopy (EIS) test, finding a maximum resistance change of 10% after cycling at 0.25C and 0.5C at 24 °C until 83% SOH. We have also included an explanation for not considering the increase in ohmic resistance during the ageing process, primarily for simplification purposes. We have added a further discussion in Section 2.4 to investigate the influence of internal resistance change. The revised assumption is highlighted in red on Page 8, point 5:

“Cell ohmic resistance changes due to ageing are not considered for the simplification purpose. The influence of internal resistance change is further discussed in Section 2.4.”

The discussion is added on Page 19:

“Over a long-term scale, a cell's internal resistance increases during the ageing process. In Raj et al.'s study [62], a maximum increase of 10% was observed in the electrochemical impedance spectroscopy (EIS) test for two groups of NCA/graphite 18650 cells, which were cycled at 0.25C and 0.5C, respectively, from an SOH of 1 to 0.83 under 24 °C. To investigate the influence of increased internal resistance on temperature distribution, a comparison between fresh cells and aged cells (with 10% increased internal resistance) is presented in Fig. S4 of the Supplementary Information. The results show that the maximum temperature difference is 0.8 °C, indicating that the temperature distribution pattern is not significantly affected by the aging process.”

8. “Before the numerical simulation, the following assumptions have been made in this study as follows: 1. (...) 3. Interconnection resistance is not considered in this study due to the focus on investigating the impact of electrical topologies on inhomogeneous ageing. In the 2D model, the connector resistance is neglected. In the 3D model, the electrical conductivity of connectors is manually set to an extremely high value of $1 \times 10^{12} \text{ Sm}^{-1}$ to minimise the voltage drop and ohmic heating across the connectors.”

The validity of this assumption is not clear for me, since in the series-parallel arrangement in the two packs considered is not the same, and therefore the interconnection resistances could play a role on inhomogeneous ageing, as shown in previous literature, e.g. in [16].

Response: The interconnection resistance is not considered in order to focus solely on the electrical connection topology, thereby representing a best-case scenario where interconnection resistance is minimal and only the effects of topology are considered. This approach helps demonstrate the level of inherent inhomogeneity that cannot be mitigated through the design of low-resistance bus bars and cell connections. We have run a sensitivity analysis on interconnection resistance with different busbar resistance and show the impact the interconnection resistance is neglectable when adopting high electric conductivity materials, such as aluminium, or hypothetical super-low resistance material (i.e. in this study)

We have added a discussion of interconnection resistance on Page 19:

“It is worth noting that, to isolate the variables, interconnection resistance is neglected in this study. However, the proposed 3D model can also assess interconnection resistance by setting the connector's electrical conductivity to its original material value. Nevertheless, the study of interconnection resistance can be case-sensitive, as it depends on the selected material's electrical properties, geometry, and joining techniques [62]. This study primarily focuses on exploring representative electrical connection topologies, and the influence of interconnection resistance is beyond the scope of this research. However, to better demonstrate the influence of interconnection resistance in this study, a sensitivity analysis is conducted by comparing current distribution at 1C discharge using three different materials: steel, aluminium, and a hypothetical super-low resistance material (i.e. in this study). The current distribution can be found in Fig. S3 of the Supplementary Information. The results indicate that current maldistribution decreases with increasing electrical conductivity; however, the difference in current distribution between aluminium and the hypothetical super-low resistance material is negligible. This also underscores the reason why aluminium is one of the most used materials

[63]. Thus, when selecting a material with high electrical conductivity (e.g. aluminium), interconnection resistance is not the primary factor influencing the inhomogeneous ageing between the straight and parallelogram connection topologies for this study.”

Note: Fig. S3 can be found at the end of this document.

Additionally, we revised assumption 2 to make it clearer, highlighted in red on Page8:

“Cell-to-cell variances due to extrinsic factors, such as interconnection resistance and welding techniques, are not accounted for in controlling the variables. This is to focus on the impact of connection topology on inhomogeneous ageing due to temperature gradients. Thus, interconnection resistance is neglected in the 2D model. In the 3D model, the electrical conductivity of connectors is manually set to an extremely high value of $1 \times 10^{12} \text{Sm}^{-1}$ to minimise the voltage drop and ohmic heating across the connectors. The influence of interconnection resistance is further discussed in Section 2.4.”

9. “Before the numerical simulation, the following assumptions have been made in this study as follows: 1. (...) 4. Heat conduction is the only heat transfer mode considered in this simulation.”

To ease understanding, the authors should clarify here if this means if what are the actual paths considered for heat transfer by conduction, e.g.: 1) cell-to-cell through the surface, 2) cell-to-cell through the busbar, 3) cell-to-cooling system, etc. I know the answer after reading the whole paper, but the authors should make clear here if they are assuming: 1) a lumped thermal cell model, i.e., if they use a volume averaged temperature, or a 2) 3D distributed thermal cell model, and reflect somewhere on what are the consequences of that in the results obtained.

Response: We have rewritten the sentence to make our statement clearer. The revised content is highlighted in red on Page 8, point 6:

“The heat transfer considered in this model is limited to heat convection (i.e. heat removed by the cooling liquid) and heat conduction (i.e. heat transfer from the cells to the cooling pipe and between cells through the busbar). Heat transfer from cell to cell and from cell to ambient air is not considered. We note cells tend to have an insulation layer in battery pack design to prevent thermal runaway propagation [44, 45].”

A diagram explaining how the 2D and the 3D model interact would be useful too, to ease understanding.

Response: 2D model focuses on investigating the impact of temperature variations on cells with fixed temperature gradient at sub-module level. 3D model focus on the inhomogeneous cycling ageing at module-level. To make the link between 2D and 3D model clearer, we updated the manuscript on Page 5, highlighted in red:

“2D Model: The 2D battery modules configured in 1P3S and 3P1S, as illustrated in Figs. 2a and 2b, are built to investigate the impact of temperature variations on cells with fixed temperature gradient at sub-module level.”

And

“3D Model: 2D models explore the inhomogeneous cycling ageing under a constant temperature difference at sub-module level. However, heat generation from cells varies during the cycling phase, indicating that the temperature difference changes during an EV’s operation [4]. Thus, a representative 3D battery module (4P6S) with two different electrical connection topologies (i.e. the straight connection topology and parallelogram connection topology) is proposed to investigate and estimate the realtime changes in temperature, current, and voltage of the cells at module level.”

Additionally, we have added 2D electrical connection schematic diagrams for the proposed module in Fig.3(a-b) to illustrate two electrical connection topologies.

Figure 3 Schematic of the representative battery module (4P6S). The schematic diagrams of electrical connection: (a) straight connection topology, and (b) parallelogram connection topology.

Also, to better illustrate the relationship between 2D and 3D models, we have reorganized the structure of ‘Section 1 Method’, by adding a subsection titled ‘1.1 Geometry Development’ on Pages 6-8.

10. Regarding other possible assumptions to be listed.

- Cell-to-cell differences are not considered, although they are well documented in the literature, e.g. in [R4]. This assumption could be fine, but this should be included in the list and the consequences of that in the results commented clearly.
- The ageing model only considers SEI growth, which might be fine too, but it should be highlighted and listed.

Response: we have added these two assumptions in bullet point 1, and 4, and highlighted in red on Page 6-8:

“1. Cell-to-cell variances due to intrinsic reasons, such as the cell capacity, internal resistance, and energy density, are not considered as this study focuses on the cell-to-cell variance in temperature-sensitive electrochemical parameters due to temperature gradients (i.e. ohmic

resistance, exchange current, and diffusion coefficient) [4, 20]. Therefore, cells are assumed to exhibit the same electrochemical properties when they are at the same temperature.

4. The ageing mechanism considered in this study is the SEI formation, which is the main ageing process in most graphite-based lithium-ion batteries [43]. Lithium plating is not considered due to it mainly occurring in low temperature or high C-rate conditions [43]."

11. Regarding the thermal model parameterization: heat capacity and thermal conductivities. Considering the type of model-based study that the authors are presenting, careful parameterization of the thermal model is mandatory.

The authors justify the cell heat capacity parameter value of 750 J/kg-K based on [46], but: 1) in that reference a lower value is provided (727 J/kg-K), 2) this is not an appropriate reference, since it is an investigation based on a fake/surrogate cell, and not a real cell, and 3) in [46] it is mentioned a wide range of possible values for heat capacity found in the literature 800-1700 J/kg-K. Authors are encouraged to find a better reference to justify the election and change the value if needed.

Regarding the axial/radial thermal conductivities, the authors justify the selection of the parameters based on reference [47], i.e. M. Al-Zareer "Numerical Study of Flow and Heat Transfer Performance of 3D-Printed Polymer-Based Battery Thermal Management." Based on that thereference too, the heat capacity of the cell is 1400 J/(kg · K), which is contradictory with previous assumptions. Moreover, in this reference, the author justifies the values of the heat capacity and thermal conductivities based on references:

- M. Doyle, T.F. Fuller, J. Newman, "Modeling of Galvanostatic Charge and Discharge of the Lithium/Polymer/Insertion Cell," J. Electrochem. Soc., 140 (6) (1993), p. 1526
- M. Doyle, "Comparison of Modeling Predictions with Experimental Data from Plastic Lithium Ion Cells," J. Electrochem. Soc., 143 (6) (1996), pp. 1890-1903.
- T.F. Fuller, M. Doyle, J. Newman, "Simulation and Optimization of the Dual Lithium Ion Insertion Cell", J. Electrochem. Soc., 141 (1) (1994), pp. 1-9.

However, in none of these references there is data about thermal parameters.

The authors are suggested to take a look to [R5] to find references to justify their selection of parameters.

Changing the values of these parameters necessarily means that the authors will have to repeat the simulations and re-write the results section. The current parameters are not valid.

Response: Thank you for providing these useful references. We have reviewed them and updated the thermal properties of the cell accordingly. All simulations have been rerun, and the simulation results have been updated to reflect these changes.

Table 1 shows the original thermal properties and the updated thermal properties of the cell.

Table 1 Thermal property of 18650 cell

	Original value	Updated value
Density kgm^{-3}	2939	2734
Heat capacity $\text{J}(\text{kgK})^{-1}$	750	830
Axial thermal conductivity $\text{W}(\text{mK})^{-1}$	30	13.35
Radial thermal conductivity $\text{W}(\text{mK})^{-1}$	1	0.78

11. The authors have implemented a 3D model in COMSOL, but the authors do not present any figure in the paper that shows actual 3D results of the temperature distribution with inhomogeneous cell temperatures. I would suggest including that, and not only 2D results plots (Fig. 7 and 8).

Response: We added a new Fig.8 (Page 22) with 3D temperature distribution included:

Figure 8 The temperature, current, voltage, and SOC distribution of the module. (a) Temperature distribution for the straight and parallelogram connection topologies at 3214 s. (b) Current distribution for the straight and parallelogram connection topologies at 600 s and 3214 s. (c) SOC distribution for the straight and parallelogram connection topologies at 3214 s. (d) Cell voltage and electrical connectors potential distribution for the straight and parallelogram connection topologies at 3214s.

We have updated Fig.9 (Page 22) with 3D SOH and temperature distribution included:

Figure 9 The SOH spread for the straight connection topology and parallelogram connection topology. (a) The 3D SOH distributions for each topology when the first cell reaches the end-of-life. (b) The parallelogram connection topology exhibits a higher ageing rate than the straight connection topology (0.1% SOH). (c) The straight connection topology exhibits a more pronounced spread of inhomogeneous cycling ageing on the cool side ($\Delta\text{SOH} = 0.5\%$) than the hot side ($\Delta\text{SOH} = 0.1\%$) due to the current maldistribution. (d) The parallelogram connection topology exhibits a more uniform spread of inhomogeneous cycling ageing on the cool side ($\Delta\text{SOH} = 0.5\%$) and the hot side ($\Delta\text{SOH} = 0.4\%$) due to a more uniform current distribution resulting from a more uniform temperature distribution within the sub-modules.

12. The key limitations and assumptions, and their impact in the results, should be commented in the conclusions – or a discussion section could be included in the paper.

Response: We have added discussions on the impacts of assumptions in Section 2 Results and discussion:

1) *We have added a discussion of interconnection resistance on Page 19:*

“It is worth noting that, to isolate the variables, interconnection resistance is neglected in this study. However, the proposed 3D model can also assess interconnection resistance by setting the connector’s electrical conductivity to its original material value. Nevertheless, the study of interconnection resistance can be case-sensitive, as it depends on the selected material’s electrical properties, geometry, and joining techniques [62]. This study primarily focuses on exploring representative electrical connection topologies, and the influence of interconnection resistance is beyond the scope of this research. However, to better demonstrate the influence of interconnection resistance in this study, a sensitivity analysis is conducted by comparing current distribution at 1C discharge using three different materials: steel, aluminium, and a hypothetical super-low resistance material (i.e. in this study). The current distribution can be found in Fig. S3 of the Supplementary Information. The results indicate that current maldistribution decreases with increasing electrical conductivity; however, the difference in current distribution between aluminium and the hypothetical super-low resistance material is negligible. This also underscores the reason why aluminium is one of the most used materials [63]. Thus, when selecting a material with high electrical conductivity (e.g. aluminium), interconnection resistance is not the primary factor influencing the inhomogeneous ageing between the straight and parallelogram connection topologies for this study.”

Note: Fig. S3 can be found at the end of this document.

2) Discussion on ageing assumption, highlighted in red on Page 23:

The ageing model only considers capacity loss due to SEI growth as it is the main ageing factor in most graphite-based lithium-ion batteries. Lithium plating is not considered, as it mainly occurs under high C-rate or low-temperature conditions, where the C-rate is under 1C, and the temperature is above 25 °C in this study. Calendar ageing rate is not considered in this study as cells experience same ageing rate when the temperature distribution is homogeneous in their resting condition at the same SOC.

Reviewer 2.

General comment:

The authors have developed 2D and 3D models to investigate the impact of serial and parallelogram connection topology architecture on inhomogeneous aging at the module level. The physical baseline of the models is well explained; however, it is unclear what are the novel contributions of the work. The overall quality of the writing and manuscript structure is good.

Response: We have updated the manuscript to emphasise where the novelty in this work lies. The revised content is highlighted in red on Page 4:

“These aforementioned studies underscore that, under identical cooling conditions, inhomogeneous ageing can be further managed by modifying electrical configurations. However, the limitations can be summarised as follows:

- 1. Most previous studies focus on parallel connections (nP1S), which may not adequately represent the current distribution at the module level.*
- 2. Most previous studies rely on 2D models, which cannot fully capture real-time temperature changes or distribution throughout the entire cycling process. In these studies, the temperature gradient is often set to a constant value (e.g. a 5 °C increment in Liu et al.’s study [18]; a fixed gradient of 12.5 °C or 25 °C in Marlow et al.’s study [17]).*
- 3. Most research has focussed on interconnection resistance and welding techniques to optimise module-level variances. To the best of the authors’ knowledge, the cell-to-cell variances caused by temperature gradients across different electrical connection topologies has not been reported.*

Therefore, it is crucial to develop a 3D battery module model that includes both parallel and series connections while considering real-time temperature changes. This will enable a comprehensive understanding of the different module-level inhomogeneous cycling ageing caused by different electrical connection topologies.”

Bullet comment:

1. The abstract fails to introduce the straight and parallelogram connection topology so that the results make sense. The authors should also rewrite the last two lines, especially because 'due to limiting the voltage range' is unclear.

Response: The reviewer makes a good point. We have added a description to the straight and parallelogram connection topology, and highlighted in red on Page 1:

"This study numerically investigates a 4P6S battery module, comprising sub-modules with two different connection topologies: 1) a straight connection topology, the sub-modules consisting of parallel-connected cells are serial connected in a linear configuration, and 2) a parallelogram connection topology, where the sub-modules are serial connected in a parallelogram configuration."

We have re-written the last part of the Abstract to make it clearer:

"We find that the straight electrical connection topology is more advantageous. This topology allows the temperature gradient to be distributed among the parallel-connected cells in the sub-modules, mitigating the issue of over(dis)charging. Consequently, it results in a higher effective capacity increase of 0.8% than the parallelogram connection topology. Additionally, it exhibits a higher State of Health (SOH) of 80.15% when the parallelogram connection topology first reaches the end-of-life (80%). However, it is noteworthy that the straight connection topology results in increased current maldistribution within sub-modules, but it is considered an acceptable trade-off."

2. The authors are recommended to focus on BTMS only to provide the baseline study reasoning. At times, the discussion seems unnecessarily engaged in the thermal gradient, BTMS, etc. Otherwise, the title of the work should be rephrased.

Response: This study investigated the combined effects of different electrical connection topologies on inhomogeneous cycling ageing due to temperature gradient stemming from the BTMS. Thus, we feel the inclusion of thermal gradient and BTMS parts are important. However, we have changed the title of the paper to make it clearer:

"Numerical investigation of variance in module-level inhomogeneous cycling ageing due to temperature gradient across different electrical connection topologies"

3. The topology type should be in a consistent format like $\alpha P\beta S$; not 15p1s (and similar on page3).

Response: We have changed "15p1s" to "15P1S" (Page 3), "11p1s" to "11P1S" (Page 3).

4. In section 1, point 1 – the authors assume a literature-driven aging rate; however, this is cell specific. Same comment for the point 2. Are these studies based on the same cell? How to justify the assumptions when the research topic is about inhomogeneous aging on the cell level?

Response: 1) Point 1: The experimental ageing data at 1C discharge and 0.5C charge under 10 °C, 25 °C, and 40 °C from Zhang et al. [55] are based on the same NCA/graphite Panasonic 18650 cell.

2) Point 2: We have updated a new literature from Raj et al. [52]. They measured the resistance change in NCA/graphite 18650 Panasonic cells using an electrochemical impedance spectroscopy (EIS) test, finding a maximum resistance change of 10% after cycling at 0.25C and 0.5C at 24 °C until 83% SOH. We have also included an explanation for not considering the increase in ohmic resistance during the ageing process, primarily for simplification purposes. We have added a further discussion in Section 2.4 to investigate the influence of internal resistance change. The revised assumption is highlighted in red on Page 8, point 5:

“Cell ohmic resistance changes due to ageing are not considered for the simplification purpose. The influence of internal resistance change is further discussed in Section 2.4.”

The discussion is added on Page 19:

“Over a long-term scale, a cell's internal resistance increases during the ageing process. In Raj et al.'s study [62], a maximum increase of 10% was observed in the electrochemical impedance spectroscopy (EIS) test for two groups of NCA/graphite 18650 cells, which were cycled at 0.25C and 0.5C, respectively, from an SOH of 1 to 0.83 under 24 °C. To investigate the influence of increased internal resistance on temperature distribution, a comparison between fresh cells and aged cells (with 10% increased internal resistance) is presented in Fig. S4 of the Supplementary Information. The results show that the maximum temperature difference is 0.8 °C, indicating that the temperature distribution pattern is not significantly affected by the aging process.”

3) We have added two new assumptions on the cell-to-cell variance (point 1 and 2, on Page 6-8):

“Point 1: Cell-to-cell variances due to intrinsic reasons, such as the cell capacity, internal resistance, and energy density, are not considered as this study focuses on the cell-to-cell variance in temperature-sensitive electrochemical parameters due to temperature gradients (i.e. ohmic resistance, exchange current, and diffusion coefficient) [4, 20]. Therefore, cells are assumed to exhibit the same electrochemical properties when they are at the same temperature.

Point 2: Cell-to-cell variances due to extrinsic factors, such as interconnection resistance and welding techniques, are not accounted for in controlling the variables. This is to focus on the impact of connection topology on inhomogeneous ageing due to temperature gradients. Thus, interconnection resistance is neglected in the 2D model. In the 3D model, the electrical conductivity of connectors is manually set to an extremely high value of $1 \times 10^{12} \text{Sm}^{-1}$ to minimise the voltage drop and ohmic heating across the connectors. The influence of interconnection resistance is further discussed in Section 2.4. The influence of interconnection resistance is further discussed in Section 2.4.”

We have also added a discussion of interconnection resistance on Page 19:

“It is worth noting that, to isolate the variables, interconnection resistance is neglected in this study. However, the proposed 3D model can also assess interconnection resistance by setting the connector's electrical conductivity to its original material value. Nevertheless, the study of interconnection resistance can be case-sensitive, as it depends on the selected material's electrical properties, geometry, and joining techniques [62]. This study primarily focuses on exploring representative electrical connection topologies, and the influence of interconnection resistance is beyond the scope of this research. However, to better demonstrate the influence of interconnection resistance in this study, a sensitivity analysis is conducted by comparing current distribution at 1C discharge using three different materials: steel, aluminium, and a hypothetical super-low resistance material (i.e. in this study). The current distribution can be found in Fig. S3 of the Supplementary Information. The results indicate that current maldistribution decreases with increasing electrical conductivity; however, the difference in current distribution between aluminium and the hypothetical super-low resistance material is negligible. This also underscores the reason why aluminium is one of the most used materials [63]. Thus, when selecting a material with high electrical conductivity (e.g. aluminium), interconnection resistance is not the primary factor influencing the inhomogeneous ageing between the straight and parallelogram connection topologies for this study.”

Note: Fig. S3 can be found at the end of this document.

3. Table A1 is mentioned to be taken from ref [32] in the text but the table itself refers to [45]. The authors should use only one reference.

Response: we have corrected the reference in Table A1 on Page 24 and highlighted it in red on Page 25.

4. The bottom views of Fig. 3d and 3e have no significance, thus, suggested to be removed.

Response: we have removed the bottom views of Fig. 3d and 3e.

5. Typo in the text on page 11 and in Fig. 4c where the cold temperature is said to be 15°C, and 10°C. Only one value should be correct, this, unfortunately, creates confusion.

Response: we have corrected the “15°C” to “10°C” in the text, highlighted in red on Page 14. We have corrected the temperature from 15°C to 10°C in Fig. 4c, highlighted in red on Page 15.

6. The theoretical models' generated influential factor calculation is made and the numbers are mentioned in the text but no physical explanations are available for many cases. The authors are encouraged to dive deeper into the aging mechanism.

Response: we have rewritten the ageing model by adding more physical explanations to each ageing factor (highlighted in red on Page 11-12):

1. f_E : High SOC values (typically resulting in high battery voltage) accelerate capacity loss [58]. f_E represents the result of either a parasitic electrochemical reduction reaction occurring on the negative electrode, or an oxidation reaction occurring on the positive electrode.
2. f_I : Battery lifetime is related to the amount of cycled equivalent full cycles. f_I represents the linear relation between the capacity fade and the number of full cycles.
3. f_{age} : Ageing history defines how many times the capacity loss rate will have been reduced when all capacity has been lost. f_{age} represents the rate of the capacity fade slowed down by the parasitic reactions, e.g. the formation of SEI.
4. f_T : The cycling ageing rate increase in both higher and lower temperature [3]. f_T is the ageing caused by temperature, described by an Arrhenius expression.

7. The model-based results are not validated in physical measurement. This is a big shortcoming. The data from the literature must be verified with the specified test for the specified topologies.

Response: The cell electrochemical and ageing model have been validated based on the experimental data. The battery module model is not validated by experiment. However, it would be difficult to undertake a good experimental validation of the full 3D model as measuring current distribution in parallel is very difficult without interfering with the system under test (i.e. shunt resistors or hall effect sensors). Additionally, we only focus on the cell-to-cell variance in temperature-sensitive electrochemical parameters (i.e. ohmic resistance, exchange current, and diffusion coefficient) due to the temperature gradient stemming from the BTMS. Other factors, such as interconnectors resistance, cell level variances can be difficult to eliminate in experimental test.

To reflect this, we have changed the title to emphasis this is a numerical investigation on the module-level inhomogeneous ageing.

We have also added a discussion of interconnection resistance on Page 19:

“It is worth noting that, to isolate the variables, interconnection resistance is neglected in this study. However, the proposed 3D model can also assess interconnection resistance by setting the connector's electrical conductivity to its original material value. Nevertheless, the study of interconnection resistance can be case-sensitive, as it depends on the selected material's electrical properties, geometry, and joining techniques [62]. This study primarily focuses on exploring representative electrical connection topologies, and the influence of interconnection resistance is beyond the scope of this research. However, to better demonstrate the influence of interconnection resistance in this study, a sensitivity analysis is conducted by comparing current distribution at 1C discharge using three different materials: steel, aluminium, and a hypothetical super-low resistance material (i.e. in this study). The current distribution can be found in Fig. S3 of the Supplementary Information. The results indicate that current maldistribution decreases with increasing electrical conductivity; however, the difference in current distribution between aluminium and the hypothetical super-low resistance material is

negligible. This also underscores the reason why aluminium is one of the most used materials [63]. Thus, when selecting a material with high electrical conductivity (e.g. aluminium), interconnection resistance is not the primary factor influencing the inhomogeneous ageing between the straight and parallelogram connection topologies for this study.”

Note: Fig. S3 can be found at the end of this document.

8. The novel contribution of the research seems also to require bigger motivation as most of the works are taken from different literature.

Response: We have summarised the research gap, and highlighted in red on Page 4:

“1. Most previous studies focus on parallel connections (nP1S), which may not adequately represent the current distribution at the module level.

2. Most previous studies rely on 2D models, which cannot fully capture real-time temperature changes or distribution throughout the entire cycling process. In these studies, the temperature gradient is often arbitrarily set to a constant value (e.g. a 5 °C increment in Liu et al.’s study [18]; fixed 12.5 °C or 25 °C gradient in Marlow et al.’s study [17]).

3. Most research has focussed on interconnection resistance and welding techniques to optimise module-level variances. To the best of the authors' knowledge, the cell-to-cell variances caused by temperature gradients across different electrical connection topologies has not been reported.

Therefore, it is crucial to develop a 3D battery module model that includes both parallel and series connections while considering real-time temperature changes. This will enable a comprehensive understanding of the different module-level inhomogeneous cycling ageing caused by different electrical connection topologies.”

9. The authors should also comment on additional module-level factors like cell-to-cell variation, pressure, electrical connection, vibration, etc. which are to be discussed as possible reasons behind inhomogeneous aging.

Response: We have added more literature on other factors that could introduce inhomogeneous aging in Introduction. We have highlighted them in red on Page 3:

“The unavoidable temperature gradient between cells can also lead to cell-to-cell variations, which can be categorised into three levels: particle level, cell level, and module level. At the particle level, defects or irregularities in electrode materials due to manufacturing techniques can lead to local inhomogeneities, impacting the performance, durability, and safety of the battery [12]. However, particle-level inhomogeneities are typically intrinsic and difficult to control. Thus, most studies focus on the cell level and explore aspects such as surface temperature inhomogeneities [13, 14], capacity variation [15], internal resistance variation [15], and mechanical stress variation [16]. Compared cell-level study, research on module-level variations is relatively less explored. Previous cell-level studies have predominantly focused on

current regulation regarding thermal gradients [17], interconnection resistance [18, 19], and welding techniques [18].

From a thermal perspective, temperature gradients lead to variances in temperature-sensitive electrochemical properties among cells, such as internal ohmic resistance, charge exchange current, and diffusion coefficient [20]. For example, ohmic resistance decreases with increasing temperature due to lithium ions migrating faster through the electrolyte [21]. The charge exchange current increases with temperature as electrodes become more reactive [22]. Similarly, the diffusion coefficient increases with temperature, enhancing the kinetics of lithium ions [23, 24]. These variations lead to uneven distribution of temperature, current, and voltage, exacerbating inhomogeneous cycling ageing in the cells [10, 25]. Liu et al. [17] found that a thermal gradient of 25°C increased aging rate by 5.2%. Various approaches have been proposed to regulate temperature homogeneity within the module, such as cell tab optimisation [26, 27], low-temperature preheating/self-heating techniques [28], battery mechanical design [29], reconfigurable battery management systems [30], and novel balancing methods [31, 32]. From the perspective of the BTMS, various aspects such as cell layouts [33, 34], flow patterns [35, 36], and BTMS geometries [37, 38] have been studied as means to improve homogeneity.”

Reviewer 3.

1. The authors investigate the unbalanced distributions among the cells with two study cases, the straight and parallelogram. The authors have done a very nice work and presented their results in a clear way.

Response: Thank you for your recognition for our work.

2. Introduction main focus is the inhomogeneities from thermal management point of view, with the various cooling methods (air-liquid, which is also not SoA). More efforts should be placed on the effect of electrical connections/ bus bars/ materials/ soldering methods etc to align with the paper title. This is also contradicting with page 3 ("this study incorporates...under changing temperatures.)

Response: 1) We have added more literature on other effects that could introduce inhomogeneous ageing in Introduction. We have highlighted them in red on Page 3:

“The unavoidable temperature gradient between cells can also lead to cell-to-cell variations, which can be categorised into three levels: particle level, cell level, and module level. At the particle level, defects or irregularities in electrode materials due to manufacturing techniques can lead to local inhomogeneities, impacting the performance, durability, and safety of the battery [12]. However, particle-level inhomogeneities are typically intrinsic and difficult to control. Thus, most studies focus on the cell level and explore aspects such as surface temperature inhomogeneities [13, 14], capacity variation [15], internal resistance variation [15], and mechanical stress variation [16]. Compared cell-level study, research on module-level variations is relatively less explored. Previous cell-level studies have predominantly focused on current regulation regarding thermal gradients [17], interconnection resistance [18, 19], and welding techniques [18].

From a thermal perspective, temperature gradients lead to variances in temperature-sensitive electrochemical properties among cells, such as internal ohmic resistance, charge exchange current, and diffusion coefficient [20]. For example, ohmic resistance decreases with increasing temperature due to lithium ions migrating faster through the electrolyte [21]. The charge exchange current increases with temperature as electrodes become more reactive [22]. Similarly, the diffusion coefficient increases with temperature, enhancing the kinetics of lithium ions [23, 24]. These variations lead to uneven distribution of temperature, current, and voltage, exacerbating inhomogeneous cycling ageing in the cells [10, 25]. Liu et al. [17] found that a thermal gradient of 25°C increased aging rate by 5.2%. Various approaches have been proposed to regulate temperature homogeneity within the module, such as cell tab optimisation [26, 27], low-temperature preheating/self-heating techniques [28], battery mechanical design [29], reconfigurable battery management systems [30], and novel balancing methods [31, 32]. From the perspective of the BTMS, various aspects such as cell layouts [33, 34], flow patterns [35, 36], and BTMS geometries [37, 38] have been studied as means to improve homogeneity.”

2) We have rewritten the sentence to make our statement clearer (highlighted in red on Page 5):

Original: This study incorporates both thermal and electrical configurations into the analysis to simulate in real-time the current and voltage distributions under changing temperatures.

Revised: It integrates thermal and electrochemical models to simulate real-time current, voltage and temperature distributions using a representative 3D battery module (4P6S).

3. Assumption 2 and 3: elaborate more on this. Why such important parameters are not considered? What are the limitations of considering only the capacity fade during ageing (and only by SEI) and how relevant is this capacity fade to electrical topology and nonuniform heat distribution? Which other ageing mechanisms the uniformities can affect?

Response: 1). The interconnection resistance is not considered in order to focus solely on the electrical connection topology, thereby representing a best-case scenario where interconnection resistance is minimal and only the effects of topology are considered. This approach helps demonstrate the level of inherent inhomogeneity that cannot be mitigated through the design of low-resistance bus bars and cell connections. We have rewritten the assumptions to make our statement clearer, and added a discussion on the influence of interconnection resistance, which can be found in the Supplementary Note 1. The revised assumptions are highlighted in red on Page 6-8, Point 1 and 2:

- 1. Cell-to-cell variances due to intrinsic reasons, such as the cell capacity, internal resistance, and energy density, are not considered as this study focuses on the cell-to-cell variance in temperature-sensitive electrochemical parameters due to temperature gradients (i.e. ohmic resistance, exchange current, and diffusion coefficient) [4, 20]. Therefore, cells are assumed to exhibit same electrochemical properties when they are at the same temperature.*

2. *Cell-to-cell variances due to extrinsic factors, such as interconnection resistance and welding techniques, are not accounted for in controlling the variables. This study focuses on examining the impact of various electrical connection topologies on inhomogeneous ageing, which is attributed to the temperature gradient stemming from the BTMS. Thus, connector resistance is neglected in the 2D model. In the 3D model, the electrical conductivity of connectors is manually set to an extremely high value of $1 \times 10^{12} \text{Sm}^{-1}$ to minimise the voltage drop and ohmic heating across the connectors. The influence of interconnection resistance is further discussed in Section 2.4.*

We recognise the cell ohmic resistance change will influence the generation. A maximum of 10% increase was observed in the NCA/graphite 18650 cell when the SOH decreased from 1 to 0.83 in Raj et al.'s study. Thus, we neglect the ohmic resistance change for the simplification purpose. We have added a further discussion in Section 2.4 to investigate the influence of internal resistance change. The revised assumption is highlighted in red on Page 6, point 5:

“Cell ohmic resistance changes due to ageing are not considered for the simplification purpose. The influence of internal resistance change is further discussed in Section 2.4.”

The discussion is added on Page 20:

“Over a long-term scale, a cell's internal resistance increases during the ageing process. In Raj et al.'s study [62], a maximum increase of 10% was observed in the electrochemical impedance spectroscopy (EIS) test for two groups of NCA/graphite 18650 cells, which were cycled at 0.25C and 0.5C, respectively, from an SOH of 1 to 0.83 under 24 °C. To investigate the influence of increased internal resistance on temperature distribution, a comparison between fresh cells and aged cells (with 10% increased internal resistance) is presented in Fig. S4 of the Supplementary Information. The results show that the maximum temperature difference is 0.8 °C, indicating that the temperature distribution pattern is not significantly affected by the aging process.”

3) We have updated ageing assumption in Point 3 and 4, on Page 8:

“Point 3: The temperature-related ageing rate varies under cycling or resting conditions, and increases at lower temperatures during cycling [41], but decreases at lower temperatures during calendar ageing [42]. This study exclusively focuses on cyclic ageing due to cells experiencing the same calendar ageing rate when the temperature distribution is homogeneous and the cell is resting.

Point 4: The ageing mechanism considered in this study is the SEI formation, which is the main ageing process in most graphite-based lithium-ion batteries [43]. Lithium plating is not considered due to it mainly occurring in low temperature or high C-rate conditions [43].”

4. Include a table with the parameters used for the electrical, thermal and ageing models and indicate their values and how they were obtained. This will increase the validity and reproducibility of the paper.

Response: A table of symbols and values used in battery and ageing mode is added in Table A3, Page 25. A table of symbols and values used in thermal mode is in Table A4, Page 26.

5. The thermal model is not clear and it needs improvements. How the surface temperature of the cells is derived? How did the thermal parameters are obtained? What is the reference temperature stated in the paper?

Response: 1) We have rewritten Section 1.2.2 Heat transfer highlighted in red on Page 12-13. We removed the equation summary table, and now explained each equation in text form.

2) The surface temperature of the cells is derived by using the energy conservation equation (Eq. (30)):

$$\rho_{batt} C_{p,batt} \partial T_{batt} / \partial t = \nabla(k_{batt} \nabla T_{batt}) + \dot{Q}_{total}$$

where the ρ_{batt} , $C_{p,batt}$, T_{batt} , and k_{batt} is the density, heat capacity and thermal conductivity of cell, respectively.

3) The thermal parameters of NCA/graphite Panasonic 18650 cell are obtained from Ref. [70-71]. The thermal parameters of other materials (e.g. cooling water, aluminium cooling pipe, steel busbar) are obtained from COMSOL in-build materials.

4) The reference temperature is 25°C. We have emphasised this by adding: “The reference temperature of this study is 25 °C.”, highlighted in red on Page 6.

6. 3D thermal model appears more as a TMS study instead of electrical connections/topology. What is the difference in the physics between the two models (straight and parallel)? How these two designs affect the heat generation of the cells or the conduction? More focus and explanations should be given on the electrical connections of the models. Do all the cells have the same parameters and states in both modules, i.e. is a single cell extrapolated to the whole module?

Response: 1) What is the difference in the physics between the two models (straight and parallel)?

We added a 2D schematic diagrams of electrical connection: (a) straight connection topology, and (b) parallelogram connection topology in Fig. 3, Page 8 to better illustrate the physical difference.

Figure 3 Schematic of the representative battery module (4P6S). The schematic diagrams of electrical connection: (a) straight connection topology, and (b) parallelogram connection topology.

2) How these two designs affect the heat generation of the cells or the conduction?

We have added a comparison of “heat generation rate” in Fig. 7 e-f, and have also added a discussion on the influence of heat generation rate for these two designs, on Page 19-20:

“Fig. 7c illustrates the heat generation rate among cells during the 1C discharge phase. The heat generation rate of a cell depends on the current and temperature, as described Eq. 23-29. It is observed that cells on the cooler side exhibit a higher heat generation rate due to experiencing greater overpotential. Additionally, a higher current maldistribution leads to a more pronounced heat generation rate gradient in the straight connection topology. However, the overall difference in heat generation between the two designs does not significantly influence the temperature distribution, as illustrated in Fig. 7a.”

Figure 7 Comparisons of module-level temperature, current, heat generation rate, SOC, and voltage change for straight and parallelogram connection topologies under 1C discharge. (a) Temperature increases from 25 °C to a maximum of 34 °C and temperature difference between two topologies is negligible (< 0.62 °C). (b) Current change shows the straight connection topology exhibits higher maldistribution due to higher temperature gradient among cells within a sub-module due to higher temperature gradient among cells within a sub-module. (c) Heat generation rate change shows cells on the cool side have higher heat generation rate due to experiencing higher overpotential, and higher current maldistribution leads to higher heat generation rate gradient. (d) SOC change shows the straight connection topology exhibits a higher SOC gradient at the end-of-discharge due to current maldistribution. (e) Voltage change shows the parallelogram connection topology exhibits a higher voltage maldistribution due to higher temperature gradient among sub-modules.

3) Do all the cells have the same parameters and states in both modules, i.e. is a single cell extrapolated to the whole module?

Cells have the same parameters if they experience the same temperature. However, when temperature changes, temperature-sensitive electrochemical parameters also change, resulting in different cell states. We updated Assumption Point 1 on Page 6 to reflect this:

1. "Cell-to-cell variances due to intrinsic reasons, such as the cell capacity, internal resistance, and energy density, are not considered as this study focuses on the cell-to-cell variance in temperature-sensitive electrochemical parameters due to temperature gradients (i.e. ohmic resistance, exchange current, and diffusion coefficient) [4, 20]. Therefore, cells are assumed to exhibit the same electrochemical properties when they are at the same temperature."

7. Is the 3D module model is validated? What is the limitations here?

Response: 1) Is the 3D module model is validated?

The battery module model has not been experimentally validated. Conducting a comprehensive experimental validation of the full 3D model is challenging, particularly in measuring current distribution in parallel configurations without interfering with the system under test, such as using shunt resistors or Hall effect sensors. Furthermore, this study specifically focuses on the cell-to-cell variance in temperature-sensitive electrochemical parameters (i.e., ohmic resistance, exchange current, and diffusion coefficient) due to temperature gradients. Other variables, such as interconnector resistance and cell level variances, can be difficult to eliminate in experimental test.

2) What is the limitations here?

We have added a discussion of interconnection resistance on Page 19:

"It is worth noting that, to isolate the variables, interconnection resistance is neglected in this study. However, the proposed 3D model can also assess interconnection resistance by setting the connector's electrical conductivity to its original material value. Nevertheless, the study of interconnection resistance can be case-sensitive, as it depends on the selected material's electrical properties, geometry, and joining techniques [62]. This study primarily focuses on exploring representative electrical connection topologies, and the influence of interconnection resistance is beyond the scope of this research. However, to better demonstrate the influence of interconnection resistance in this study, a sensitivity analysis is conducted by comparing current distribution at 1C discharge using three different materials: steel, aluminium, and a hypothetical super-low resistance material (i.e. in this study). The current distribution can be found in Fig. S3 of the Supplementary Information. The results indicate that current maldistribution decreases with increasing electrical conductivity; however, the difference in current distribution between aluminium and the hypothetical super-low resistance material is negligible. This also underscores the reason why aluminium is one of the most used materials [63]. Thus, when selecting a material with high electrical conductivity (e.g. aluminium), interconnection resistance is not the primary factor influencing the inhomogeneous ageing between the straight and parallelogram connection topologies for this study."

Note: Fig. S3 can be found at the end of this document.

8. Fig. 5c and 5f. Why the Vohmic increases for the 35degC, while it decreases for the 15degC? Please indicate the charge-discharge profile with respect to time.

Response:

1) Fig. 5c and 5f. Why the Vohmic increases for the 35degC, while it decreases for the 15degC?

From Eq. 3, $\eta_{ohm} = R_{ohm}I_{batt}$, we can see that η_{ohm} is dependent on R_{ohm} and I_{batt} . In a serial connection, I_{batt} is the same for every cell; thus, η_{ohm} depends solely on R_{ohm} . However, in a parallel connection, a cell with higher temperature will undergo a higher I_{batt} due to lower resistance. At the same time, R_{ohm} is lower. When these two factors are multiplied, they result in the corresponding change in η_{ohm} .

2) Please indicate the charge-discharge profile with respect to time.

We have added charge-discharge profile to the Fig. 5, on Page 16:

Figure 5 Comparison of electrochemical parameters for serial and parallel connections. The variances in (a-b) current, (c) ohmic resistance, (d-e) ohmic overpotential, (f) dimensionless exchange current, (g-h) activation overpotential, (i) time constant and (j-k) concentration overpotential under 15 °C, 25 °C, and 35 °C.

9. Section 2.4, what is the natural convection (no cooling temperature behavior) of the cells/modules for both cases?

Response: The natural air convection is not considered in this model as it is not the main cooling mechanism, and the cells tend to have an insulation layer in battery pack design to

prevent thermal runaway propagation [53,54]. We have updated Assumption Point 6 (Page 6) to make the heat transfer mechanism clearer:

“Point 6. The heat transfer considered in this model is limited to heat convection (i.e. heat removed by the cooling liquid) and heat conduction (i.e. heat transfer from the cells to the cooling pipe and between cells through the busbar). Heat transfer from cell to cell and from cell to ambient air is not considered. We note cells tend to have an insulation layer in battery pack design to prevent thermal runaway propagation [44, 45].”

10. How is the balancing of the BMS could affect the conclusions of this paper? What is the cost of each module with respect to the components used?

Response: 1) How is the balancing of the BMS could affect the conclusions of this paper?

We have added a discussion on the potential impact of BMS on active balancing, and highlighted in red on Page:23:

“Current battery management system (BMS) commonly estimates the SOC of individual cells in series by measuring voltage or coulomb counting. However, for individual cells connected in parallel (i.e. the sub-module), the BMS treats them as a single 'lumped' cell due to the lack of access to specific currents and temperatures of each cell [64]. This approach prevents the BMS from differentiating the aging scenario within the sub-module. The BMS could attempt to compensate for inhomogeneous aging by having series-connected cells at different voltages, but this would limit the operating capacity of the pack due to voltage limit constraints.”

2) What is the cost of each module with respect to the components used?

Cost is not considered in this study due to it beyond the scope of this study. However, regarding the cost difference between the two designs, it primarily lies in the electrical connectors. We believe this will not significantly impact the overall cost.

Supplementary Information

Figure S 1. Comparison of temperature distribution for the fresh and aged cells.

Figure S3. Comparison of the variance in current distribution for different interconnection materials. (a) Steel. (b) Aluminium. (c) Hypothetical super-low resistance.

REVIEWERS' COMMENTS:

Reviewer #2 (Remarks to the Author):

Following the response to the general comment, the authors should look at the recently published paper if it has addressed some of the shortcomings - <https://doi.org/10.1038/s44172-023-00153-5>

This reviewer is still not convinced with the responses to Points 7 and 8. The novel contribution is weak unless an experimental verification of the 3D outcome is demonstrated. The named concerns could be highlighted in case of significant deviation from the experimental study.

Reviewer #4 (Remarks to the Author):

This work provides some numerical investigation on the cell-to-cell variance due to temperature gradient. The paper is well written. I have two main concerns:

1. The results of the numerical investigation are only valid if the model is correct and can be experimentally validated, which is unfortunately not the case in this work.
2. The lack of experimental validation also leads to the three highlights mentioned in the answer to Reviewer 2's 8th comment being unconvincing. Because these highlights are hard to be claimed when the model used in this work has not been validated.

To summarize, I really recommend the authors carry out further experimental validations of the whole model to convince the readers of the main conclusions of this work.

Reviewer #2 (Remarks to the Author):

1. Following the response to the general comment, the authors should look at the recently published paper if it has addressed some of the shortcomings - <https://doi.org/10.1038/s44172-023-00153-5>

This reviewer is still not convinced with the responses to Points 7 and 8. The novel contribution is weak unless an experimental verification of the 3D outcome is demonstrated. The named concerns could be highlighted in case of significant deviation from the experimental study.

Response: The model has been validated at the cell level. Experimental validation at the module level can be challenging, as other factors mentioned in the manuscript (i.e., internal resistance, cell capacity variety, and energy density variety) are difficult to eliminate in the experiment. In this work, we focus solely on one factor, i.e. electrical connection topology. Thus, isolating this single factor and validating it via experimental work is difficult. However, We do recommend conducting experimental work in the future to analyse inhomogeneous aging, taking all these factors into consideration together. The revised content is highlighted in red on Page 21:

“The future work will focus on module-level experimental validation to further examine the current maldistribution and inhomogeneous ageing within the proposed battery module.”

We also reviewed the recommended reference, and cited the results from this work to support our conclusion. The revised content is highlighted in red on Page 20:

“In this study, if the discharge stops when the first cell reaches the cut-off voltage, the parallelogram connection topology has an effective discharge capacity of 88.6%, while the straight connection topology has an effective discharge capacity of 89.4% at a 1C discharge, resulting in a 0.8% capacity difference. This find agrees with Marlow et al.’s study¹² that within parallel-connected cells, accessible capacity is reduced due to the end-of-discharge SOC deficit.”

Reviewer #4 (Remarks to the Author):

This work provides some numerical investigation on the cell-to-cell variance due to temperature gradient. The paper is well written. I have two main concerns:

1. The results of the numerical investigation are only valid if the model is correct and can be experimentally validated, which is unfortunately not the case in this work.

Response: The model has been validated at the cell level. Experimental validation at the module level can be challenging, as other factors mentioned in the manuscript (i.e., internal resistance, cell capacity variety, and energy density variety) are difficult to eliminate in the experiment. In this work, we focus solely on one factor, i.e. electrical connection topology. Thus, isolating this single factor and validating it via experimental work is difficult.

2. The lack of experimental validation also leads to the three highlights mentioned in the answer to Reviewer 2's 8th comment being unconvincing. Because these highlights are hard to be claimed when the model used in this work has not been validated.

To summarize, I really recommend the authors carry out further experimental validations of the whole model to convince the readers of the main conclusions of this work.

Response: We do recommend conducting future experimental work to analyse inhomogeneous aging, taking all these factors into consideration together. The revised content is highlighted in red on Page 21:

"The future work will focus on module-level experimental validation to further examine the current maldistribution and inhomogeneous ageing within the proposed battery module."